# HIF1α stabilization in hypoxia is not oxidant-initiated

Amit Kumar[1,2,3], Manisha Vaish[1,4], Saravanan S Karuppagounder[1,2,3], Irina Gazaryan[5], John W Cave[1,2,3], Anatoly A Starkov[2,3], Elizabeth T Anderson[6], Sheng Zhang[6], John T Pinto[7], Austin M Rountree[8], Wang Wang[9], Ian R Sweet[8], Rajiv R Ratan[1,2,3]*

[1]Burke Neurological Institute, White Plains, New York, United States; [2]Brain and Mind Research Institute, Weill Medical College of Cornell University, New York, United States; [3]Department of Neurology, Weill Medical College of Cornell University, New York, United States; [4]Pandemic Response Lab, New York, United States; [5]Department of Anatomy and Cell Biology, New York Medical College, New York, United States; [6]Institute for Biotechnology, Cornell University, Ithaca, United States; [7]Department of Biochemistry and Molecular Biology, New York Medical College, Valhalla, United States; [8]Department of Medicine, University of Washington, Seattle, United States; [9]Department of Pain and Anesthesiology, University of Washington, Seattle, United States

*For correspondence:
rrr2001@med.cornell.edu

Competing interest: The authors declare that no competing interests exist.

**Abstract** Hypoxic adaptation mediated by HIF transcription factors requires mitochondria, which have been implicated in regulating HIF1α stability in hypoxia by distinct models that involve consuming oxygen or alternatively converting oxygen into the second messenger peroxide. Here, we use a ratiometric, peroxide reporter, HyPer to evaluate the role of peroxide in regulating HIF1α stability. We show that antioxidant enzymes are neither homeostatically induced nor are peroxide levels increased in hypoxia. Additionally, forced expression of diverse antioxidant enzymes, all of which diminish peroxide, had disparate effects on HIF1α protein stability. Moreover, decrease in lipid peroxides by glutathione peroxidase-4 or superoxide by mitochondrial SOD, failed to influence HIF1α protein stability. These data show that mitochondrial, cytosolic or lipid ROS were not necessary for HIF1α stability, and favor a model where mitochondria contribute to hypoxic adaptation as oxygen consumers.

## Introduction

Over the past decade, our ability to monitor and manipulate reactive oxygen species (ROS) has grown enormously. These technological advances provide a novel view on how ROS interact with cells to modulate function. Specifically, ROS such as peroxide can act as cellular messengers. Messenger functions for ROS reflect their tight spatial control within cells. The tight spatial control of ROS has enabled their critical roles in growth factor signaling, inflammation, and regeneration (*Jain et al., 2013*; *Lei and Kazlauskas, 2014*; *Hameed et al., 2015*). Specific signaling roles for ROS are facilitated by the existence of motifs in proteins such as phosphatases that render these proteins specifically susceptible to redox modulation (*Bae et al., 1997*; *Lee et al., 1998*; *Salmeen et al., 2003*).

An important area of biology where ROS signaling has been highly investigated but where no consensus has emerged is hypoxic adaptation. Seminal work from Semenza, Kaelin, and Ratcliffe demonstrated that when oxygen tension falls below a critical threshold, that a family of enzymes dependent on oxygen, iron and 2-oxoglutarate known as the HIF Prolyl Hydroxylases (HIF PHDs) decrease their activity, leading to diminished hydroxylation of the alpha subunit of HIF transcription factors (*Semenza*

*and Wang, 1992*; *Wang and Semenza, 1993*; *Wang et al., 1995*; *Maxwell et al., 1999*; *Epstein et al., 2001*; *Ivan et al., 2001*; *Ivan et al., 2002*). Diminished HIF1α hydroxylation decreases recruitment of a key E3 Ubiquitin ligase, the von Hippel Lindau (VHL) protein, which allows HIF1α to avoid proteasomal degradation. Stabilized HIF1α dimerizes with its constitutively active partner to bind to hypoxia response elements in a coordinate gene cassette that leads to hypoxic adaptation at a cellular, local, and systemic level (*Wang et al., 1995*; *Wood et al., 1996*; *Jiang et al., 1996*).

Since the mitochondrial electron transport chain (ETC) generates reactive oxygen species via electrons that leak from ETC components, bind to oxygen to produce superoxide anions, mitochondrial ROS production is tightly linked with availability of oxygen. Previous studies have shown that increasing levels of HIF1α stability in response to progressively lower oxygen concentrations required other factors besides oxygen to regulate HIF PHD activity (*Chandel et al., 2000*; *Brunelle et al., 2005*; *Mansfield et al., 2005*; *Bell et al., 2007*). Using then state of the art tools to monitor ROS, it was shown that during hypoxia, ROS were increasing and that pharmacological tools that nullified this increase in ROS diminished HIF1α stability (*Chandel et al., 2000*). A compelling model emerged that hypoxia increases the flux of electrons via Rieske Iron Sulfur cluster proteins in complex III and this leads to an increase in mitochondrial ROS generation via the ubiquinone binding site near the outer aspect of the inner mitochondrial membrane (the Qo site) (*Bell et al., 2007*). Peroxide generated at this site could then diffuse through the outer mitochondrial membrane to inhibit HIF PHDs via either direct redox modulation of HIF PHDs (*Bell et al., 2007*) or activation of established redox sensitive p38 MAP kinase signaling (*Emerling et al., 2005*) .

Since its inception, the role of mitochondria established by Chandel and Schumacker in HIF signaling has been validated (*Agani et al., 2000*; *Mansfield et al., 2005*; *Taylor, 2008*). However, an alternate ROS-independent view of how mitochondria regulate HIF1α stability in hypoxia has also been advanced. Pharmacological inhibition of ETC complexes or genetic knockdown of Rieske Fe/S proteins or cytochrome c not only inhibits mitochondrial ROS production (*Brunelle et al., 2005*; *Chandel et al., 2000*; *Mansfield et al., 2005*; *Bell et al., 2007*), it also inhibits oxygen consumption. In this distinct scheme, inhibition of ETC function would alter the kinetics of availability of cytosolic oxygen levels. Indeed, several groups have provided data supporting the importance of respiratory chain driven mitochondrial oxygen consumption in dictating cellular oxygen gradients. These gradients have been hypothesized to decrease oxygen concentrations to levels required to inhibit enzymatic activity of oxygen dependent HIF PHDs-the key upstream regulator of HIF1α stability (*Doege et al., 2005*; *Hagen et al., 2003*; *Chua et al., 2010*). There are a number of pathological conditions such as cerebral ischemia, acute kidney disease, ischemic heart disease, neonatal hypoxia-ischemic brain injury, cancer, and pulmonary arterial hypertension where both HIF1α and ROS have been reported to be important players (*Chen et al., 2011*; *Chen et al., 2012*; *Weidemann et al., 2008*; *Tomsa et al., 2019*; *Semenza, 2014*; *Kibel et al., 2020*; *Sheldon et al., 2014*; *Qin et al., 2019*; *Semenza, 2012*; *Sosa et al., 2013*; *Bryant et al., 2016*; *Demarco et al., 2010*). However, the molecular interplay between ROS and HIF1α is still poorly defined. For instance, in ischemic stroke , ischemia-reperfusion injury leads to an increase in both ROS and HIF1α. While HIF PHD inhibitors, which stabilize HIF1α and activate its downstream pathway, have shown significant benefit in providing neuroprotection, decreasing ROS levels with antioxidants (which would decrease HIF1α signaling if we consider ROS as an upstream regulator of HIF1α stability) has also shown benefits from cerebral ischemia (*Green and Shuaib, 2006*). Thus, understanding the role of ROS in HIF1α-mediated adaptation could guide more precise ways to manipulate this pathway for therapeutic benefit (*Weidemann et al., 2008*; *Conde et al., 2012*; *Semenza, 2014*; *Sheldon et al., 2014*; *Semenza, 2012*; *Bryant et al., 2016*).

In this manuscript, we leverage a host of complementary approaches that support the conclusion that peroxide is dispensable in regulated HIF1α stability in hypoxia. Unexpectedly, our results suggest that HIF1α stability in hypoxia is not oxidant-initiated.

## Results
### Mitochondria are involved in hypoxic HIF1α stabilization in SH-SY5Y human neuroblastoma cells

In the current study, we leveraged a HIF-luciferase reporter as a more quantitative biomarker of oxygen-dependent changes in endogenous HIF1α protein stabilization in hypoxia. We first established the sensitivity and dynamic range of HIF-luciferase reporter. The HIF-luciferase reporter contains the

oxygen dependent domain (ODD) of HIF1α fused to luciferase (ODD-luciferase). We have previously demonstrated that this reporter behaves like endogenous HIF but does not influence endogenous HIF activity (*Smirnova et al., 2010*; *Karuppagounder et al., 2013*). Here, we validated the reporter using adaptaquin (AQ), which we have previously established as a specific HIF PHD inhibitor (*Karuppagounder et al., 2016*). We found that endogenous HIF1α protein levels could be monitored dynamically across a wide range of AQ concentrations (*Figure 1A and B*). As an independent measure, we also assayed the activity of HIF1α-luciferase reporter by measuring luciferase activity. These assays showed that the HIF1α-luciferase reporter measured by photometric luciferase activity possesses a high dynamic range and low coefficient of variation (*Figure 1C*). Moreover, we also assessed change in endogenous HIF transcriptional activity by measuring changes in the mRNA levels of HIF target genes, *Bnip3* and *Eno2*. We found dose dependent dynamic changes in the mRNA levels of HIF target genes, *Bnip3* and *Eno2* in response to increasing concentrations of AQ (*Figure 1D and E*). Indeed, changes in HIF1α-luciferase activity measurements with increasing doses of AQ showed a strong correlation with quantitative changes in endogenous HIF1α protein levels measured by quantitative immunoblotting (*Figure 1F*) as well as quantitative changes in endogenous mRNA levels of HIF target genes, *Bnip3* and *Eno2* (*Figure 1G and H*). These findings confirmed exogenous ODD-luciferase activity as a sensitive and specific reporter of endogenous HIF1α.

Mitochondria have been previously shown to be involved in hypoxic HIF1α stabilization in different cell lines (*Chandel et al., 2000*; *Chua et al., 2010*; *Yang et al., 2012*). We wanted to verify the necessity of mitochondria in SH-SY5Y human neuroblastoma cells. To this end, we treated SH-SY5Y cells with different mitochondrial electron transport chain (ETC) inhibitors such as rotenone (ETC complex I inhibitor), myxothiazol (ETC complex III inhibitor), antimycin A (ETC complex III inhibitor), and sodium azide (ETC complex IV inhibitor) and exposed cells, in parallel, to normoxia or hypoxia for 4 hr and, thereafter, assessed ODD-luciferase activity in cell lysates. We found a dose-dependent decrease in ODD-luciferase activity in response to increasing concentrations of all of the ETC complex inhibitors in hypoxic cells (*Figure 1I–L*). These findings confirmed mitochondrial involvement in hypoxic HIF1α stabilization in SH-SY5Y cells.

## Antioxidant enzymes are not homeostatically induced in hypoxia

Increases in ROS that are sufficient for signaling or toxicity can trigger homeostatic transcriptional increases in antioxidant enzymes (*Christman et al., 1989*). To assess whether hypoxia results in similar homeostatic increases in antioxidant protein expression, we exposed human neuroblastoma (SH-SY5Y) cells to hypoxia for 8 hours and measured protein expression levels of peroxisomal, cytosolic and mitochondrial antioxidant enzymes including catalase, glutathione peroxidase-1 (GPX1), glutathione peroxidase-4 (GPX4), MnSOD, and Peroxiredoxin-3 (PRDX3) (*Figure 2A*). The 8 hr time point was chosen to monitor homeostatic changes in antioxidant enzymes because this would be 6 hr following observable HIF1α stability in hypoxia in SH-SY5Y cells, which should provide adequate time for homeostatic increases to initiate transcriptional or post-transcriptional adaptations. At the 8 - hr time point, the protein level of the peroxisomal antioxidant catalase, and antioxidants present in both cytosol and mitochondria, such as GPX1 and GPX4, showed no change protein levels in response to hypoxia (*Figure 2B and C*). However, since hypoxia induces HIF1α-dependent mitophagy (*Aminova et al., 2008*; *Zhang et al., 2008*), mitochondrial mass is decreased with increasing duration of hypoxia, including decreases in mitochondrial DNA and proteins. Accordingly, we normalized distinct mitochondrially targeted antioxidant enzymes to the level of citrate synthase, a mitochondrial protein. When normalized to citrate synthase, the expression levels of mitochondrial antioxidants such as MnSOD and PRDX3 also did not change in hypoxia (*Figure 2D and E*). To establish whether these findings apply to non-transformed cells, we studied expression levels of antioxidant enzymes in hypoxia in primary neurons. Similar to neuroblastoma cells, antioxidant enzyme levels did not change in hypoxia in post-mitotic neurons (*Figure 2F–I*). Although 8 hr hypoxic exposure did not show changes in various antioxidants, it is possible that levels of antioxidants might have changed before 8 h time point. In order to explore this possibility, we assessed changes in above mentioned antioxidants for different time points such as 2 , 4 , and 8 hr in SH-SY5Y ODD-Luc cells. However, we did not see changes in the protein levels of these antioxidants at any time point (*Figure 2—figure supplement 1*). Alternative possibilities for not observing changes in antioxidant levels could be not looking at right antioxidant enzymes or that changes in ROS levels are too small to induce detectable

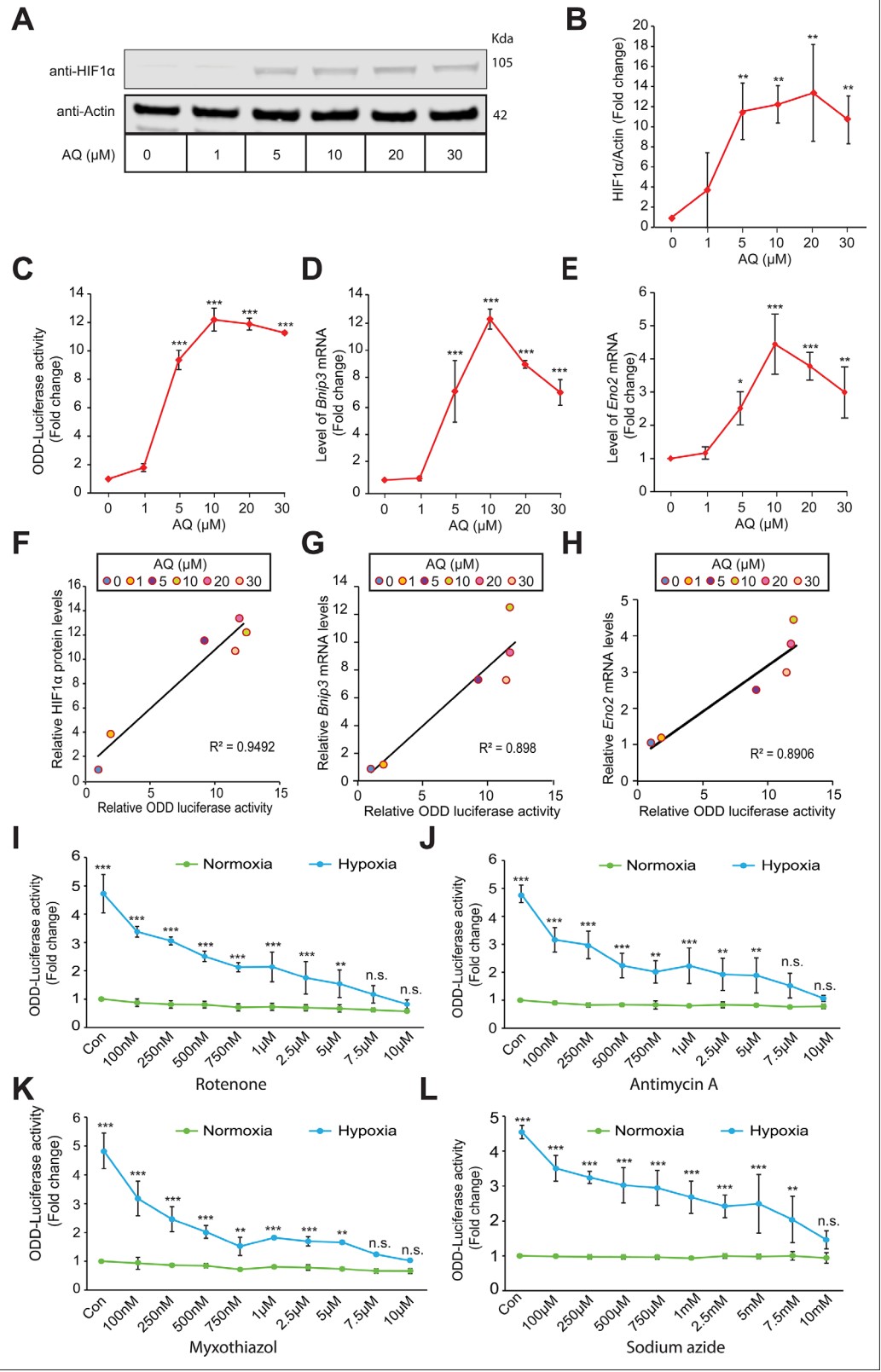

**Figure 1.** Mitochondrial electron transport chain is involved in hypoxic HIF1α stabilization in human neuroblastoma cells. Experiments A - H establish specificity and sensitivity of HIF-reporter activity (i.e. ODD-Luciferase activity) while experiments I-L establish involvement of mitochondrial ETC in the hypoxic HIF1α stabilization. SH-SY5Y ODD-Luc. cells that stably overexpressed ODD (HIF **O**xygen **D**ependent **D**omain-luciferase

*Figure 1 continued on next page*

*Figure 1 continued*

fusion) were treated with increasing concentrations of adaptaquin (AQ, 1 µM – 30 µM) for 4 hr and were, thereafter, processed for assessment of either changes in HIF protein level via immunoblotting (**A, B**) or changes in HIF reporter luciferase activity (i.e. ODD-luciferase activity) (**C**) or changes in expression of HIF1α target genes, *Eno2* and *Bnip3* (**D, E**). (**F**) Correlation between relative changes in HIF reporter luciferase activity (i.e. ODD-luciferase activity) and relative changes in HIF1α protein. (**G, H**) Correlation between relative changes in HIF reporter luciferase activity (i.e. ODD-luciferase activity) and relative changes in HIF1α target genes, *Eno2* and *Bnip3*. (**I–L**) SH-SY5Y ODD-Luc. cells were treated with increasing concentrations of rotenone (Complex I inhibitor), antimycin A (Complex III inhibitor), myxothiazol (Complex III inhibitor), and sodium azide (Complex IV inhibitor) and then were exposed to either normoxia or hypoxia for 4 hr. Thereafter, HIF reporter luciferase activity was measured in cell lysates. All experiments were performed in three independent sets. The statistical analyses were performed using One way ANOVA with Dunnett post-test (**B–E**) and Student's t test (**J–L**). (**B–E**) (n.s.), (*), (**), and (***) indicate non-significant difference and the statistical differences of $p < 0.05$, $p < 0.01$, and $p < 0.001$ with respect to normoxia control. Two-way ANOVA with Bonferroni's post-test was used for all statistical analyses (**I–L**). (n.s.), (**), and (***) indicate non-significant difference or statistical differences of $p < 0.01$ and $p < 0.001$ with respect to respective control or treatment concentration under normoxia.

The online version of this article includes the following source data for figure 1:

**Source data 1.** Original western blot files for *Figure 1A*.

---

homeostatic changes in these antioxidant enzymes. Together, these indirect measures suggest that possible increase in ROS levels during hypoxia are less likely. To further confirm if ROS levels increase in hypoxia or not, we directly measured real time changes in ROS levels in hypoxia by using a sensitive ROS marker, HyPer.

## Peroxide levels do not increase during hypoxia

HyPer is a fusion protein composed of the peroxide-sensitive domain of the prokaryotic transcription factor, OxyR, and yellow fluorescent protein that is a reporter for cellular peroxide (*Belousov et al., 2006*). This reporter is not only sensitive and specific, but its activity is also ratiometric, which factors out differences in fluorescence due to cell geometry, path length, and reporter concentration. We have previously shown that enhanced pH buffering of the extracellular medium alleviates the putative effects of acidic pH during hypoxia on the reporter, and that fluorescent ratios can be calibrated in cells to known peroxide concentrations (*Neal et al., 2016*).

To monitor changes in the level of hydrogen peroxide ($H_2O_2$), we used real-time imaging of rat pancreatic islet cells in hypoxia with strong pH buffering. Pancreatic islet cells, rather than neurons or neuron-like cells, were used for initial analysis because mitochondrial peroxide in pancreatic islet cells increases in response in extracellular glucose levels, as measured using HyPer imaging. Elevated glucose, therefore, can be used as a positive control for mitochondrial peroxide increases. Additionally, elevated glucose increases insulin release in beta islet cells, which enables insulin release assays to establish cell viability if there is an absence of peroxide changes during hypoxia. Similar real-time measures of cell function are not readily available in primary neurons or SH-SY5Y neuroblastoma cells. Prior studies have also established that glucose-induced peroxide formation is derived from the mitochondria. Specifically, increased expression of mitochondrial catalase decreased glucose-induced HyPer signals (catalase scavenges peroxide), as did decreased expression of mitochondrial SOD (MnSOD; the enzyme that converts superoxide to peroxide) (*Neal et al., 2016*). Finally, signaling levels of peroxide measured by HyPer are 1/20th those of peroxide levels required for toxicity (*Neal et al., 2016*), which demonstrates the sensitivity of peroxide measurements using HyPer in this cell type. Before assaying peroxide in hypoxia, we verified that 2 hr of hypoxia induced the established HIF1α target genes *Kdm6b* and *Bnip3* in islet cells in our flow culture system (*Figure 2J and K*; *Choudhry and Harris, 2018*). This time point was selected to evaluate the role of peroxide in mediating the earliest changes in HIF1α stability in hypoxia because it is well before mitochondrial autophagy is induced (*Figure 2*).

Accordingly, based on the sensitivity and specificity of the reporter assay, we were confident that mitochondrial peroxide was measurable in islet cells if it were increased in hypoxia. $H_2O_2$, NAD(P)H, and insulin secretion rates were measured simultaneously as a function of glucose concentration for 2 hr under hypoxic conditions (1% $O_2$) (*Figure 2L*). Increasing glucose concentration from 3 mM to 20mM elicited expected increases in peroxide, NADPH levels, and insulin secretion rates under

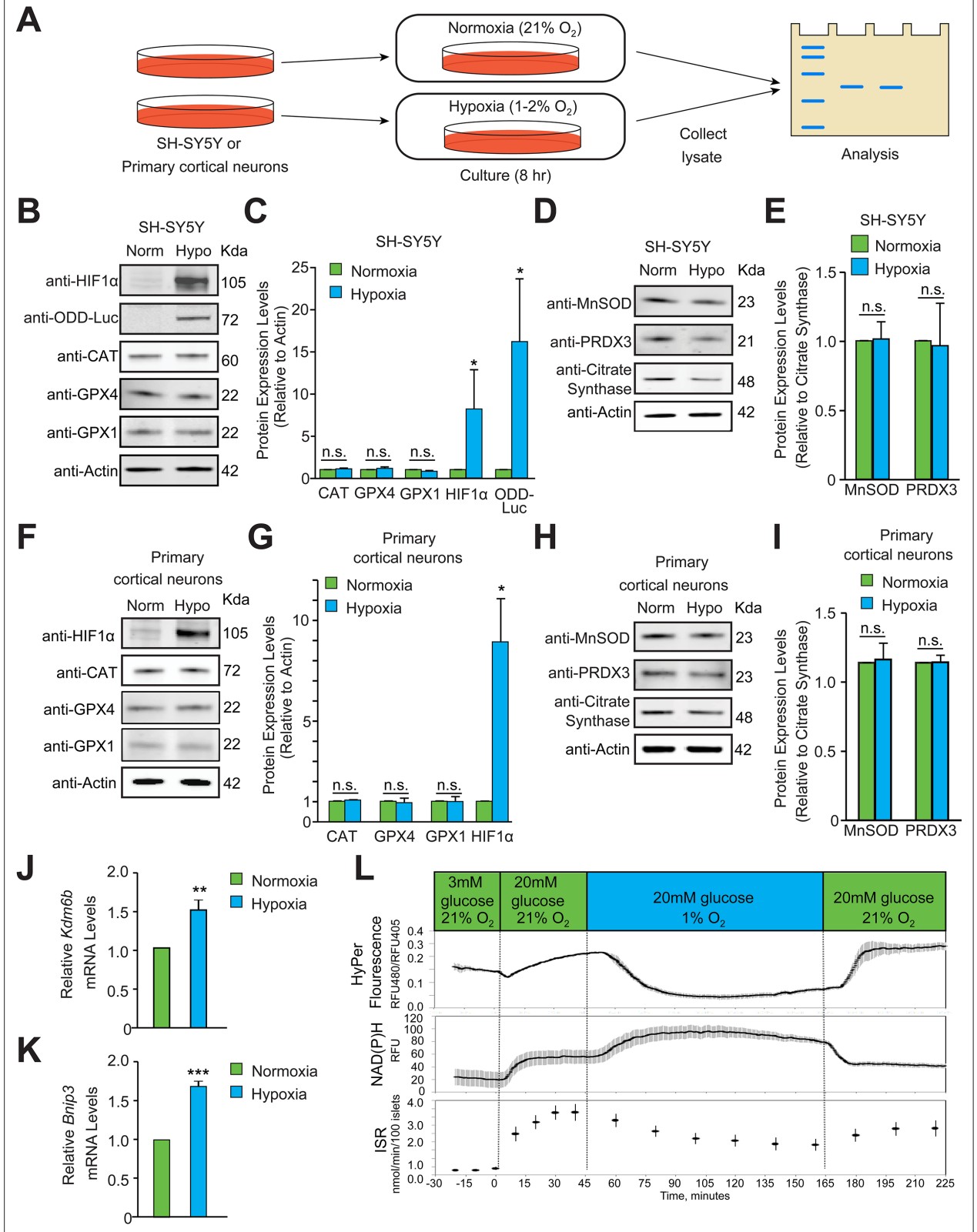

**Figure 2.** Hypoxia does not induce homeostatic increases in antioxidant enzymes or increase peroxide levels. (**A**) Experimental approach employed to examine changes in the protein levels of endogenous antioxidant enzymes in hypoxia. Immunoblots or densitometric analyses of catalase (peroxisome), GPX1 and GPX4 (cytosol and mitochondria) or MnSOD and PRDX3 (mitochondria) in SH-SY5Y ODD Luc. cells that stably overexpressed ODD (HIF Oxygen Dependent Domain-luciferase fusion) (**B–E**) or PCNs (**F-I**) exposed to normoxia or hypoxia for 8 hr. The protein levels of catalase, GPX4, and

*Figure 2 continued on next page*

*Figure 2 continued*

GPX1 were normalized to actin while those of MnSOD and PRDX3 were normalized to the mitochondrial protein, citrate synthase. A monoclonal antibody to luciferase (indicated as anti-ODD-Luc. in the figure) was used to detect changes in ODD-luciferase protein levels in SH-SY5Y ODD-Luc. cells. (**J–K**) Pancreatic islets were exposed to hypoxia for 2 hr and then were lysed and processed for mRNA expression analysis of HIF1α target genes, *Kdm6b* and *Bnip3*. The densitometric data and gene expression data were pooled from three independent experiments in the form of mean ± SD. The statistical analyses of densitometric data and gene expression data were done using Student's t test (**B–K**). (n.s.) indicates non-significant difference, (**\*\***) indicates $p < 0.01$ and (**\*\*\***) indicates $p < 0.001$ with respect to respective normoxia controls. (**L**) Hypoxia leads to large decrease in $H_2O_2$ levels (top), increased NAD(P)H (middle), and decreased insulin secretion (bottom). Glucose stimulation by 20 mM glucose was added as reference, and oxygen levels were changed using an artificial gas equilibration device placed inline in the flow system. All three experiments were carried out separately but using the same flow culture system.

The online version of this article includes the following source data and figure supplement(s) for figure 2:

**Source data 1.** Original western blot files for *Figure 2B, D*.

**Source data 2.** Original western blot files for *Figure 2F, H*.

**Figure supplement 1.** Hypoxic exposure for different time periods does not induce homeostatic increases in antioxidant enzymes.

**Figure supplement 1—source data 1.** Original western blot files for *Figure 2—figure supplement 1*.

**Figure supplement 2.** Hypoxia does not increase peroxide levels in SH-SY5Y cells or Hep3B cells.

normoxia. When islet cells under the same glucose concentrations (20mM) were exposed to 1% oxygen, $H_2O_2$ decreased by more than 80%. This reduction occurred in the midst of increased NAD(P)H levels. NADP(H) likely increased due to diminished utilization by the mitochondrial electron transport chain. Glucose-stimulated insulin secretion decreased by about 50% in response to hypoxia, but remained well above unstimulated rates, indicating that islets remained functional throughout the study. Indeed, restoring steady state $O_2$ levels (20%) resulted in expected increases in peroxide and insulin secretion with concomitant decreases in NAD(P)H, which indicates hypoxia delivered under the conditions of our experiments is not toxic to islet cells (*Figure 2L*).

To establish the generalizability of these findings to other cell types, we measured HyPer reporter fluorescence ratios in hypoxic conditions in SH-SY5Y and Hep3B human hepatocellular carcinoma cells. Ratiometric imaging of both cell types showed no change in peroxide levels following 2 hr of hypoxia (*Figure 2—figure supplement 2C and F*), which was sufficient time to activate HIF1α-dependent gene expression (*Figure 2—figure supplement 2A and B*, 2D, and 2E). This absence of changes in peroxide levels during hypoxia could not be attributed to a lack of HyPer reporter responsiveness to peroxide in these cell types since exogenous addition of peroxide (*Figure 2—figure supplement 2G*) following the hypoxic exposure led to expected, significant increases in reporter activity. Together with our antioxidant protein expression results, these data suggest that mitochondrial peroxide either decreases or is unchanged by hypoxia in primary (pancreatic beta islets) and transformed cell types (SH-SY5Y, Hep3B), respectively.

## HIF1α stabilization is not oxidant-initiated in hypoxia

Prior studies showed that decreasing ROS by forced expression of individual antioxidant enzymes can decrease HIF1α protein levels (*Brunelle et al., 2005*; *Chandel et al., 2000*). To confirm these findings with our HIF1α-luciferase reporter, we forced expression of either catalase (a peroxide scavenger), GPX1 (a peroxide scavenger), or MnSOD (a superoxide scavenger and peroxide generator) (*Figure 3A*). SH-SY5Y cells expressing ODD-Luc were transduced with individual antioxidant enzymes encoded in distinct adenoviral constructs or an adenovirus encoding GFP only as a protein control. Seventy-two hours following infection, GFP expression was observed in nearly 90 % of SH-SY5Y cells (*Figure 3—figure supplement 1*). Accordingly, cells were tested for specific enzyme activities of MnSOD, Catalase or GPX1. These studies showed seven-, eleven-, and threefold increase in specific activity over GFP controls for MnSOD-, Catalase-, and GPX1-expressing cells, respectively (*Figure 3—figure supplement 2A and C*). Moreover, using a cell death assay with DNA binding Sytox blue dye (dead cell stain), we also confirmed the activities of these antioxidants under normoxia and hypoxia by indirectly assessing the protection conferred by these antioxidants from increasing concentrations of exogenous $H_2O_2$ treatment. As expected, MnSOD did not improve protection as compared to GFP control while catalase and GPX1 overexpression significantly enhanced protection under both normoxia and hypoxia (*Figure 3—figure supplement 2D and F*).

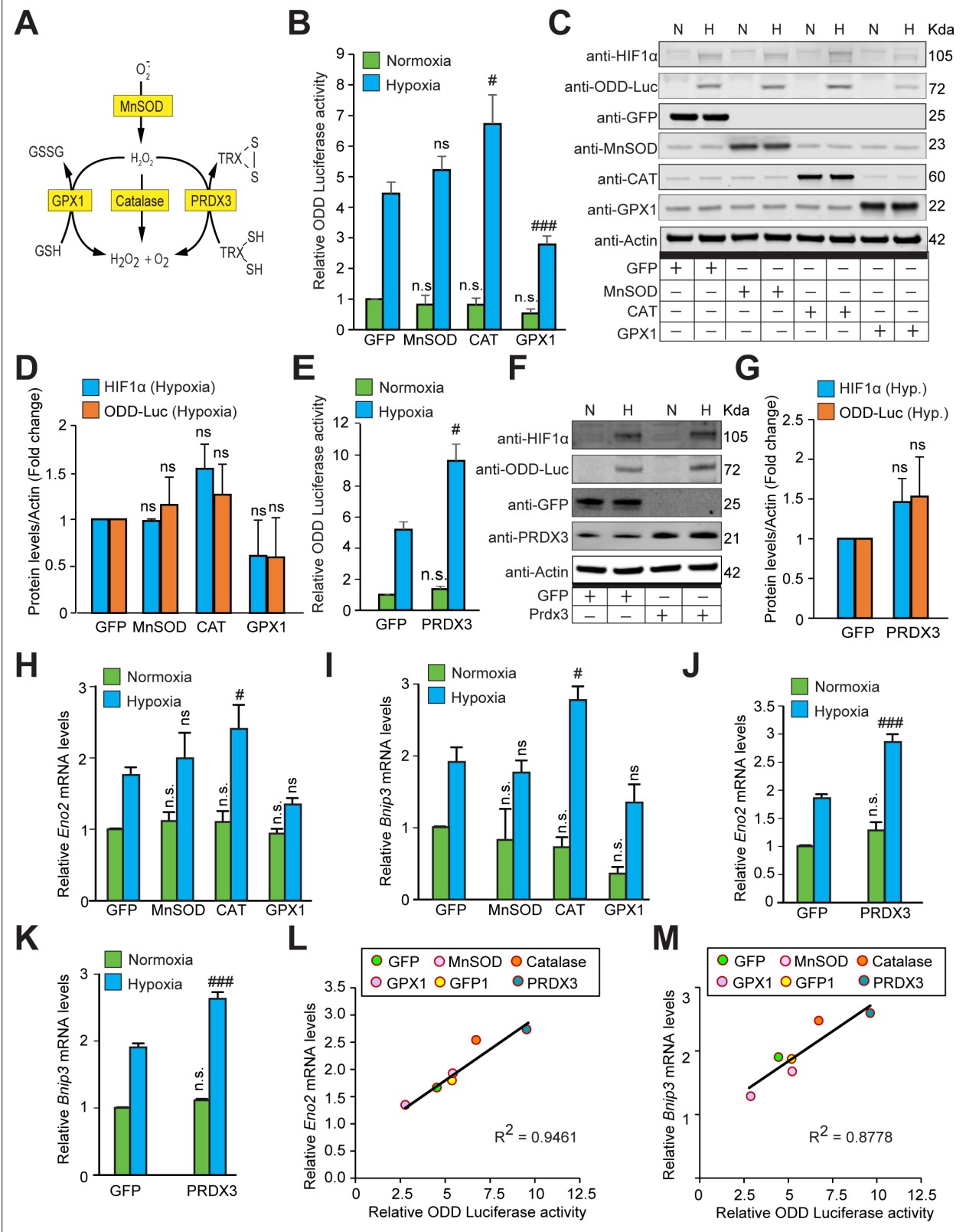

**Figure 3.** The stabilization of HIF1α and its transcriptional activity are not oxidant-initiated in hypoxia. (**A**) A schematic diagram showing known mechanisms of $H_2O_2$ detoxification by peroxisomal, cytosolic and mitochondrial antioxidants. (**B–K**) SH-SY5Y cells stably expressing ODD-luciferase were transduced with adenoviruses encoding distinct antioxidant enzymes for 72 hr and then exposed to normoxia or hypoxia in parallel and were either processed for luciferase activity assay (A measure of quantitative changes in ODD) (**B and E**) or immunoblot analysis (A measure of quantitative

*Figure 3 continued on next page*

*Figure 3 continued*

changes in protein levels of ODD and HIF1α) (C, D, F, and G) or gene expression analyses of HIF1α target genes, *Eno2* and *Bnip3* (**H–K**). (**L, M**) Correlation between relative changes in either *Eno2* or *Bnip3* and relative ODD-luciferase activities. Two-way ANOVA with Bonferroni post-test was used for comparisons in B, E, H, I, J, and K. One-way ANOVA with Dunnett post-test was used in the statistical analysis in D while Student t test in was used in G. (n.s.) indicates non-significant difference with respect to GFP control under normoxia while (ns), (#) and (###) indicate non-significant difference and the statistical differences of $p < 0.05$, and $p < 0.001$, respectively, with respect to GFP control in hypoxia. All experiments were performed as three independent sets and a representative blot or an average of three independent sets was shown in the figure. "N" stands for normoxia and "H" stands for hypoxia.

The online version of this article includes the following source data and figure supplement(s) for figure 3:

**Source data 1.** Original western blot files for *Figure 3C*.

**Source data 2.** Original western blot files for *Figure 3F*.

**Figure supplement 1.** Validation of transgene expression after forced expression of transgenes encoded within adenoviral vectors in SH-SY5Y cells.

**Figure supplement 2.** Validation of the enzyme activities of various antioxidants in normoxia and hypoxia.

**Figure supplement 3.** Validation of the functional activity of antioxidants to reduce reactive oxygen species.

**Figure supplement 4.** The stabilization of HIF1α is not oxidant-initiated in hypoxia in primary cortical neurons.

**Figure supplement 4—source data 1.** Original western blot files for *Figure 3—figure supplement 4C*.

**Figure supplement 4—source data 2.** Original western blot files for *Figure 3—figure supplement 4E*.

**Figure supplement 5.** The stabilization of HIF1α is not oxidant-initiated in hypoxia in HeLa cells.

**Figure supplement 5—source data 1.** Original western blot files for *Figure 3—figure supplement 5E*.

**Figure supplement 5—source data 2.** Original western blot files for *Figure 3—figure supplement 5E*.

**Figure supplement 5—source data 3.** Original western blot files for *Figure 3—figure supplement 5I*.

**Figure supplement 6.** The stabilization of HIF1α is not oxidant-initiated in hypoxia in Hep3B cells.

**Figure supplement 6—source data 1.** Original western blot files for *Figure 3—figure supplement 6A*.

**Figure supplement 6—source data 2.** Original western blot files for *Figure 3—figure supplement 6E*.

**Figure supplement 6—source data 3.** Original western blot files for *Figure 3—figure supplement 6I*.

As a final test of whether overexpression of these adenoviral constructs of antioxidants decrease oxidant production in SH-SY5Y cells or not, we measured 5, 6 carboxydichlorofluorescein fluorescence (a non-selective redox sensitive reporter) using flow cytometry. These experiments confirmed the ability of GPX1, PRDX3, and catalase to decrease steady state DCF oxidation, presumably resulting from oxidants generated physiologically (*Figure 3—figure supplement 3A and B*). However, there was no significant change in peroxide level in response to MnSOD expression despite a significant increase in the MnSOD enzyme activity and protein level. This was unexpected and could reflect compensatory activation of other antioxidants such as GPX1, GPX4, or Prdx3 in response to increased MnSOD activity. We verified that DCF loading and corresponding antioxidant effects were not different from normoxic or hypoxic cells, arguing against the possibility that our redox reporter or the antioxidant enzymes are behaving differently in normoxia and hypoxia (*Figure 3—figure supplement 3C and D*). Importantly, Chandel and other groups had added DCF before exposing cells to hypoxia. But there was a concern that if changes in DCF signal are because of differential accumulation of DCF in hypoxic cells as compared to normoxic cells or because of an increased oxidant signaling. To minimize the possibility of increased DCF accumulation during hypoxic exposure, we added DCF after exposing cells to normoxia/hypoxia. With this protocol, we found decrease in DCF signal in control GFP expressing cells with hypoxic exposure compared to control GFP expressing cells with normoxic exposure. Additionally, we did not find an increase in DCF signaling with $H_2O_2$ treatment in cells, which were exposed to hypoxia. This could be because of either decrease in ROS level during hypoxia, which likely increased with $H_2O_2$ to the level of GFP control under normoxia but didn't go further up or because of increased scavenging capacity of SH-SY5Y cells under hypoxia.

Having verified increases in individual antioxidant activities (MnSOD, GPX1, and Catalase) of overexpression antioxidant constructs, we examined the effect of these manipulations on HIF1α stability. The mitochondrial peroxide model of HIF regulation predicts that MnSOD should increase HIF1α stability in hypoxia, whereas GPX1 and catalase should diminish HIF1α stability. In contrast to

these predictions, we found that MnSOD had no effect on HIF1α luciferase reporter activity, while Catalase increased reporter activity, and GPX1 decreased it (*Figure 3B*). To verify that our HIF1α luciferase reporter accurately reflected endogenous HIF1α levels, we performed quantitative fluorescence immunoblotting. These assays showed changes in endogenous HIF1α protein are similar to those identified with the HIF1α reporter (*Figure 3C and D*). From these findings, we conclude that HIF1α levels are not correlated with mitochondrial peroxide production. Since catalase is a peroxisomal enzyme and GPX1 localizes to the cytosol and mitochondria, our results could not formally exclude the possibility that GPX1 localization to the mitochondria allowed it to decrease HIF1α levels, while Catalase's inability to penetrate this compartment did not allow it to decrease HIF1α levels. To address this possibility, we forced expression of PRDX3, a member of the peroxiredoxin family of antioxidant enzymes that functions as a thioredoxin-dependent peroxide reductase in the mitochondria. Contrary to GPX1, PRDX3 increased levels of HIF1α in hypoxia as measured by either HIF1α luciferase reporter activity (*Figure 3E*) or quantitative fluorescence immunoblotting (*Figure 3F and G*). We also confirmed these findings in primary neurons (*Figure 3—figure supplement 4C-F* ), where *Figure 3—figure supplement 4* adenoviral constructs also effectively increased expression of the antioxidant enzymes (*Figure 3—figure supplement 4A and B*).

Since HIF1α is a transcription factor, we next asked if peroxides play indispensable role in HIF-dependent transcription by examining *Eno2* and *Bnip3* mRNA expression levels, which are two established HIF1α target genes (*Aminova et al., 2005*; *Poitz et al., 2014*), in neuroblastoma SH-SY5Y cells. As expected, catalase and PRDX3 overexpression led to significant increase in the expression of both *Eno2* and *Bnip3* mRNA levels while GPX1 overexpression decreased the expression of both genes and MnSOD overexpression did not significantly change the expression of either of these genes (*Figure 3H–K*). Moreover, strong correlation observed between changes in HIF dependent transcription as assessed by changes in both HIF1α target genes and HIF1α protein levels as assessed by HIF1α luciferase reporter levels (*Figure 3L and M*) further confirmed that peroxides are dispensable for both hypoxic HIF1α protein stability and its transcriptional activity.

Our findings did not exclude the possibility that antioxidants alter HIF1α protein levels by differentially regulating either *Hif1a* mRNA synthesis or *Hif1a* mRNA stability. Accordingly, we monitored *Hif1a* mRNA levels in the cells overexpressing Catalase, GPX1, and PRDX3. Quantitative PCR revealed that *Hif1a* mRNA levels were not changed in a manner that would contradict the observed ROS-independent changes in HIF1α stability (*Figure 4A and B*). To confirm that changes observed in HIF1α protein stability are related to changes in its half-life, we examined the stability of HIF1α protein in the presence of cycloheximide, which suppresses de novo protein synthesis in cells pre-treated either with or without MG132. As expected, we found that Catalase and PRDX3, which increase HIF1α protein levels, also increased HIF1α half-life (*Figure 4C, D and E*). By contrast, GPX1, which diminished HIF1α protein levels, decreased HIF1α half-life (*Figure 4D*). Moreover, MG132 treatment significantly enhanced Hif-1α half-life in all cases showing that the antioxidant led changes in hypoxic Hif-1α stabilization were the results of alterations in proteasomal degradation.

To establish whether our findings in primary neurons and neuroblastoma cells can be extended to non-neural cell types, we examined the ability of antioxidant enzymes capable of modulating peroxide levels to modulate HIF1α stability in hypoxic Hep3B hepatocarcinoma cells and hypoxic HeLa cervical cancer cells. Forced expression of Catalase, GPX1, or PRDX3 using adenoviral vectors significantly increased in protein levels of each of the antioxidant enzymes in HeLa or Hep3B cells (*Figure 3—figure supplements 5 and 6*). Similar to primary neurons or neuroblastoma cells, we did not observe a uniform reduction in HIF1α stability in hypoxia by enzymes whose common activity is to decrease peroxide in these non-neural cell types (*Figure 3—figure supplements 5 and 6*). Altogether, our findings suggest that peroxide levels are uncoupled from HIF1α stability in neural and non-neural cells.

## Neither reactive oxygen species nor reactive lipid species regulate HIF-1α stability in hypoxia

Previous compelling evidence showed that reactive lipid species (RLS) are sufficient to drive HIF-dependent transcription via their effects on FIH inhibition without affecting HIF1α stability (*Masson et al., 2012*). Accordingly, we forced expression of GPX4, a selenoprotein that neutralizes RLS (*Figure 5A*). To manipulate steady-state levels, we used GPX4 protein fused to an optimized destabilization domain (dd) from the prokaryotic dihydrofolate reductase gene. The dd domain destabilizes

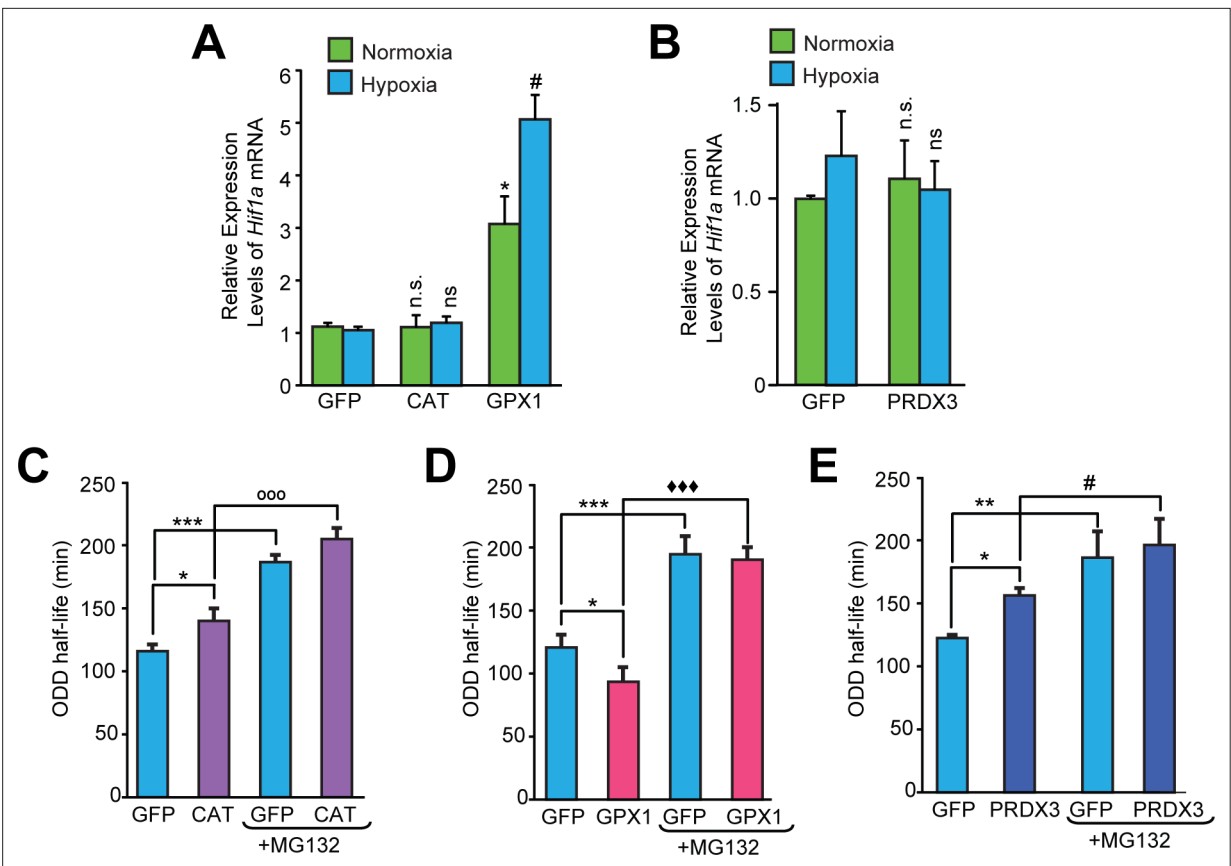

**Figure 4.** Divergent changes in HIF1α protein levels induced by antioxidant enzymes capable of scavenging peroxide cannot be attributed to differential changes in *Hif1a* mRNA synthesis or stability. (**A, B**) Relative changes in *Hif1a* mRNA in SH-SY5Y cells in response to forced expression of various antioxidant enzymes. Data were pooled from three independent experiments in the form of mean ± SD. One-way ANOVA with Dunnett's post-test was used for comparing cells expressing catalase or GPX1 with respect to cells expressing GFP and Student's t test was used for comparing cells expressing PRDX3 with respect to GFP. (n.s.) indicates non-significant difference with respect to respective GFP controls under normoxia while (ns) and (#) indicate non-significant difference and the statistical difference of p < 0.05, respectively, with respect to GFP control in hypoxia. (**C, D, E**) Changes in half-life of HIF1α in SH-SY5Y ODD-Luc cells expressing catalase or GPX1 or PRDX3 with respect to that of respective GFP controls in hypoxia. ODD Half-life was assessed by performing a pulse chase experiment by adding 35 µM cycloheximide at every 20 min for a total of 4 hr using luciferase activity assay in SH-SY5Y cells expressing these antioxidants pre-treated either with or without 10 µM MG132. Data were pooled from three independent experiments in the form of mean ± SD. Two-way ANOVA with Bonferroni's post-test was used for statistical analysis. (*), (**), and (***) indicate statistical differences of p < 0.05, p < 0.01, and p < 0.001 with respect to respective GFP controls in hypoxia. (°°°), (♦♦♦) and (#) represent statistical differences of p < 0.001 w.r.t CAT, p < 0.001 w.r.t GPX1 and p < 0.05 w.r.t PRDX3, respectively.

GPX4 protein unless trimethoprim (TMP, 10 µM) is present (*Figure 5B*). TMP enhanced GPX4 levels in neuroblastoma cells expressing ddGPX4 but increasing GPX4 levels had no effect on hypoxia-induced HIF1α stability (*Figure 5C–E*) or HIF1α-dependent transcription (*Figure 5F, G*). We verified that GPX4 diminished RLS (*Figure 5—figure supplement 1*) and neutralized ferroptosis induced by glutamate, a form of cell death mediated by reactive lipid species that is abrogated by GPX4 (*Figure 5H and I*; *Tan et al., 1998*). Altogether, these findings argue against a central role for hydrogen peroxide or lipid peroxides in mediating HIF1α stabilization in hypoxia.

## Discussion

Seminal studies have supported the notion that mitochondria are essential regulators of hypoxic adaptation possibly acting via their ability to generate peroxide (*Agani et al., 2000*; *Chandel et al., 1998*). In this paper, we show that HIF1α stability mediated by HIF PHDs during hypoxia does not require peroxide. These data include our inability to detect an increase in peroxide during hypoxia (*Figure 2L*); the lack of homeostatic changes in antioxidant protein expression during

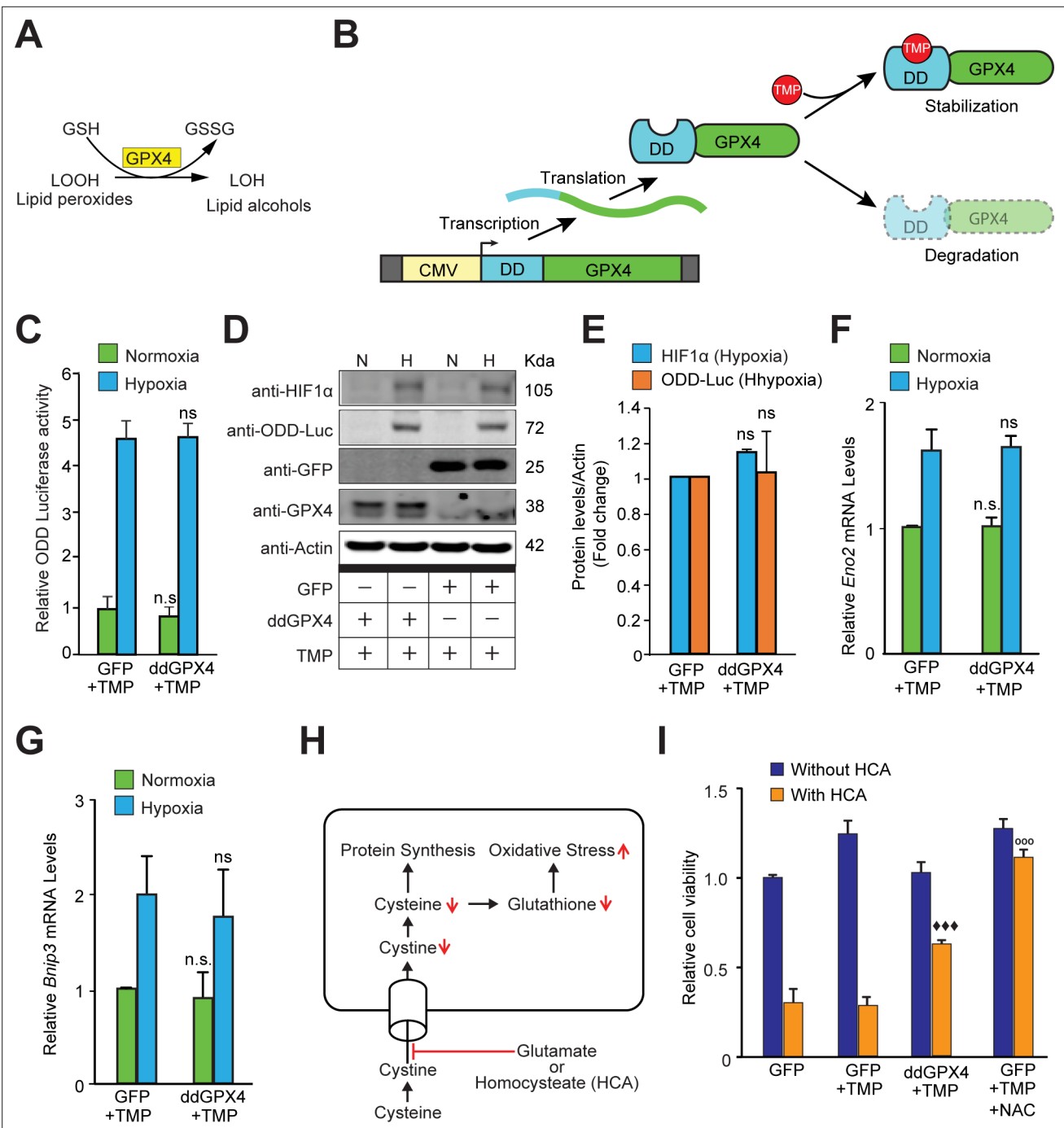

**Figure 5.** Reactive lipid species do not regulate HIF1α stabilization or its transcriptional activity in hypoxia. (**A**) A schematic diagram showing that GPX4 converts lipid hydroperoxides into lipid alcohols using GSH as a cofactor. (**B**) A schematic showing regulated protein expression of GPX4 fusion containing an optimized destabilization domain when exposed to the antibiotic, Trimethoprim (TMP). Reversible stability of GPX4 protein was conferred by fusing its coding sequence to a destabilization domain sequence (mutants of *E. coli* dihydrofolate reductase). Accordingly, GPX4 protein possessing the destabilization domain is degraded resulting in low steady levels of GPX4. Trimethoprim binds to and neutralizes the destabilization domain stabilizing GPX4 protein in a dose-dependent manner. (**C–E**) SH-SY5Y ODD-Luc cells were transduced with adenoviral vectors encoding a destabilized form of GPX4 (ddGPX4) or GFP and then exposed to normoxia/hypoxia and were either processed for luciferase activity assay (**C**) or immunoblotting (**D**) and densitometric quantitation of three independent western blots of HIF1α and ODD-Luc protein were shown in (**E**). 10 µM TMP was added to ddGPX4 expressing cells after 60 h of adenoviral incubation for 12 hr to achieve stabilized GPX4 expression. (**F, G**) Relative changes in mRNA levels of HIF1α target genes, *Eno2* and *Bnip3* in SH-SY5Y cells. (**H**) A schematic diagram depicting the glutathione depletion model of oxidative stress. (**I**) PCNs were transduced with GFP/GPX4 for 24 hr and then treated with 5 mM HCA. Cells were simultaneously treated with 10 µM TMP. Then, cells were incubated

*Figure 5 continued on next page*

*Figure 5 continued*

for 24 hr to induce oxidative stress. Thereafter, viability of cells was measured via the MTT assay. 100 µM NAC was used as positive control. The final values were pooled as mean ± S.D. of three independent experiments. Two-way ANOVA with Bonferroni's post-test was used for statistical analyses. One-way ANOVA with Dunnett's post-test was used for comparing DCF fluorescence and viability. (n.s.), (*) and (***) indicate non-significant difference, and the statistical differences of $p < 0.05$, and $p < 0.001$, respectively, with respect to GFP control under normoxia while (ns), (#) and (###) indicate non-significant difference, and the statistical differences of $p < 0.05$, and $p < 0.001$, respectively, with respect to GFP control in hypoxia. (°°°) indicates statistical difference of $p < 0.001$ with respect to GFP treated with HCA only while (♦♦♦) indicates statistical difference of $p < 0.001$ with respect to GFP treated with both TMP and HCA.

The online version of this article includes the following source data and figure supplement(s) for figure 5:

**Source data 1.** Original western blot files for *Figure 5D*.

**Figure supplement 1.** Validation of the functional activity of GPX4 to reduce reactive lipid species.

hypoxia (*Figure 2B* and *Figure 2—figure supplement 1I*); and the failure of forced expression of antioxidant enzymes (Catalase, GPX1, and PRDX3) with the common ability to diminish cellular peroxide to influence HIF1α stability and transcription in the same direction (*Figure 3*). Our findings agree with prior studies that showed: (1) that HIF PHDs are not inhibited by exogenously added peroxide (*Chua et al., 2010*), and (2) that forced expression of an alternative oxidase which directly transfers electrons from coenzyme Q to oxygen to form water maintains HIF1α stability in hypoxia, despite reducing superoxide generation at Complex III (*Chua et al., 2010*). Our results cannot be attributed to differences in mitochondrial ROS generation by transformed versus primary cells, or to differences in neuron-like versus non-neural cells, as SH-SY5Y neuroblastoma cells and primary cortical neurons showed similar effects, as did Hep3B hepatocellular carcinoma cells and HeLa cervical cancer cells. While we cannot exclude the possibility that culture conditions such as serum lots could reconcile our results with prior studies, in aggregate, the findings favor a role for mitochondria in modulating HIF1α stability via their effects as oxygen consumers not as peroxide second messenger generators.

## Peroxide scavengers have distinct effects on hypoxic HIF1α stability

Prior studies have shown that in some cell types, ROS generation in hypoxia could be related to increased oxidant production or decreased defenses (*Naranjo-Suarez et al., 2012*). In order to address that the imbalance of oxidants and antioxidants plays the regulatory role in mediating hypoxia signaling, we forced expression of distinct antioxidant enzymes known to either decrease (GPX1, Catalase, and PRDX3) or enhance peroxide levels (MnSOD). Despite evidence for increased activity of the antioxidant enzymes studied using multiple experimental approaches, we found that HIF1α stability and transcription did not correlate with effects on peroxide levels in neuroblastoma cells and primary neurons (*Figure 3* and *Figure 3—figure supplement 4*). Similar uncoupling was observed in non-neural cell types as well (*Figure 3—figure supplements 5 and 6*). The results uncouple peroxide generation in the mitochondria from HIF1α stability via the HIF PHDs. It is likely that the GPX1, catalase, and PRDX3 influenced HIF1α stability either via indirect but distinct effects on oxygen consumption, or alternatively via differential but direct effects on proteins that influence HIF1α regulation.

## HIF1α stability under hypoxia is not regulated by reactive lipid species (RLS)

Recent studies have highlighted the potential role that RLS could play in regulating HIF transcription rather than HIF1α stability. Accordingly, we forced expression of GPX4, a selenoprotein with known ability to neutralize reactive lipid species. Despite being active in combating RLS-mediated ferroptotic death in transformed cells (*Figure 5I*), GPX4 had no effect on HIF1α stability in hypoxia (*Figure 5C–E*) or HIF-dependent transcription (*Figure 5F and G*). These data suggest that while some RLS (e.g. tert-butyl hydroperoxide) are sufficient to activate HIF transcription, they are not necessary to stabilize HIF1α or drive HIF1α-dependent transcription in hypoxia.

# Materials and methods

**Key resources table**

| Reagent type (species) or resource | Designation | Source or reference | Identifiers | Additional information |
|---|---|---|---|---|
| Antibody | Anti-GFP (Rabbit polyclonal) | Cell Signaling Technology | Cat#2555; RRID:AB_390710 | WB (1:1000) |
| Antibody | Anti-MnSOD (Rabbit polyclonal) | Sigma-Aldrich | Cat#HPA001814; RRID:AB_1080134 | WB (1:10000) |
| Antibody | Anti-Catalase (Mouse monoclonal) | Sigma-Aldrich | Cat#C0979; RRID:AB_258720 | WB (1:1000) |
| Antibody | Anti-GPX1 (Rabbit monoclonal) | Cell Signaling Technology | Cat#3,286 S; RRID:AB_2067572 | WB (1:1000) |
| Antibody | Anti-GPX1 (Rabbit polyclonal) | Novus Biologicals | Cat#NBP1-33620; RRID:AB_10004091 | WB (1:2000) |
| Antibody | Anti-GPX4 (Rabbit polyclonal) | LSBio | Cat#LS B1596 | WB (1:1000) |
| Antibody | Anti-PRDX3 (Rabbit polyclonal) | Novus Biologicals | Cat#NBP2-19777 | WB (1:2000) |
| Antibody | Anti-Luciferase (Mouse monoclonal) | Santa Cruz Biotechnology | Cat#sc-74548; RRID: AB_1125118 | WB (1:1000) |
| Antibody | Anti-HIF1α (Rabbit polyclonal) | Novus Biologicals | Cat#NB100-479; RRID:AB_10000633 | WB (1:1000) |
| Antibody | Anti-HIF1α (Mouse monoclonal) | Novus Biologicals | Cat#NB100-105; RRID:AB_10001154 | WB (1:500) |
| Antibody | Anti-Citrate synthase (Rabbit monoclonal) | Cell Signaling Technology | Cat#14,309 S; RRID:AB_2665545 | WB (1:3000) |
| Antibody | Anti-β-actin (Rabbit polyclonal) | Sigma-Aldrich | Cat#A2066; RRID:AB_476693 | WB (1:10000) |
| Antibody | Anti-β-actin (Mouse monoclonal) | Sigma-Aldrich | Cat#A5316; RRID:AB_476743 | WB (1:10000) |
| Other | Ad-GFP | ViraQuest, Inc (North liberty, IA) | MSRN: 22,576 | Adenoviral construct |
| Other | Ad-ddGPX4 | ViraQuest, Inc (North liberty, IA) | MSRN: 22,665 | Adenoviral construct |
| Other | Ad5CMV-GFP | Univ. of Iowa, Viral Vector Core Facility | | Adenoviral construct |
| Other | Ad5CMV-MnSOD | Univ. of Iowa, Viral Vector Core Facility | | Adenoviral construct |
| Other | Ad5CMV-CAT | Univ. of Iowa, Viral Vector Core Facility | | Adenoviral construct |
| Other | Ad5CMV-GPX1 | Univ. of Iowa, Viral Vector Core Facility | | Adenoviral construct |
| Other | Ad-CMV-GFP | Vector Biolabs | | Adenoviral construct |
| Other | Ad-h-PRDX3 | Vector Biolabs | ADV-219772 | Adenoviral construct |
| Other | Ad-h-GRX1 | Vector Biolabs | ADV-209995 | Adenoviral construct |
| Other | Ad-HyPer-cyto vector | Vector Biolabs | | Adenoviral construct |
| Biological sample (*Rattus norvegicus*) | Islets from Sprague–Dawley male rats | Charles River | | |
| Biological sample (*Mus musculus*) | Primary cortical neurons from CD-1 strain mice | Charles River | | |

*Continued on next page*

*Continued*

| Reagent type (species) or resource | Designation | Source or reference | Identifiers | Additional information |
|---|---|---|---|---|
| Chemical compound, drug | Adaptaquin | Tocris Bioscience | 5953; CAS Number 385786-48-1 | |
| Chemical compound, drug | Rotenone | Sigma-Aldrich | R8875; CAS Number 83-79-4 | |
| Chemical compound, drug | Antimycin A | Sigma-Aldrich | A8674; CAS Number 1397-94-0 | |
| Chemical compound, drug | Myxothiazol | Sigma-Aldrich | T5580; CAS Number 76706-55-3 | |
| Chemical compound, drug | Sodium azide | Sigma-Aldrich | S2002; CAS Number 26628-22-8 | |
| Chemical compound, drug | Potassium cyanide (KCN) | Sigma-Aldrich | 207810; CAS Number 151-50-8 | |
| Chemical compound, drug | FCCP | Sigma-Aldrich | C2920; CAS Number 370-86-5 | |
| Chemical compound, drug | Trimethoprim (TMP) | Sigma-Aldrich | T7883; CAS Number 738-70-5 | |
| Chemical compound, drug | L-Homocysteic acid (HCA) | Sigma-Aldrich | H9633; CAS Number 14857-77-3 | |
| Chemical compound, drug | *N*-acetyl-L-cysteine (NAC) | Sigma-Aldrich | A7250; CAS Number 616-91-1 | |
| Chemical compound, drug | Hydrogen peroxide solution, 30% w/w | Sigma-Aldrich | H1009; CAS Number 7722-84-1 | |
| Commercial Assay or kit | Rat Insulin Radioimmunoassay (RIA) | Millipore-Sigma | RI-13K | |
| Commercial Assay or kit | DCFDA-Cellular Reactive Oxygen Species Detection Assay Kit | Abcam | ab113851 | |
| Commercial Assay or kit | DC Protein Assay Kit 1 | Bio-Rad | 5000111 | |
| Commercial Assay or kit | Luciferase Assay System | Promega | E1501 | |
| Commercial Assay or kit | MTT assay (CellTiter 96 Non-Radioactive Cell Proliferation Assay) | Promega | G4100 (G4101+ G4102) | |
| Other | Sytox blue nucleic acid stain | ThermoFisher Scientific | SS11348 | |
| Cell line (*Homo sapiens*) | SH-SY5Y | ATCC | CRL-2266; RRID:CVCL_0019 | |
| Cell line (*Homo sapiens*) | HeLa | ATCC | ATCC CCL-2; RRID:CVCL_0058 | |
| Cell line (*Homo sapiens*) | Hep3B | ATCC | ATCC HB-8064; RRID:CVCL_0326 | |
| Sequence-based reagent | FAM labeled *Enolase2* (Taqman probe) | ThermoFisher Scientific | Hs00157360_m1 | |
| Sequence-based reagent | FAM labeled *Bnip3* (Taqman probe) | ThermoFisher Scientific | Hs00969291_m1 | |
| Sequence-based reagent | FAM labeled *Hif1α* (Taqman probe) | ThermoFisher Scientific | Hs00153153_m1 | |
| Sequence-based reagent | VIC labeled *human β actin* (Taqman probe) | ThermoFisher Scientific | 4326315E | |
| Sequence-based reagent | FAM labeled *RNA28S5* (Taqman probe) | ThermoFisher Scientific | Hs03654441_s1 | |
| Sequence-based reagent | VIC labelled *RNA28S5* (Taqman probe) | ThermoFisher Scientific | Hs03654441_s1 | |
| Sequence-based reagent | FAM labeled *Bnip3* (Taqman probe) | ThermoFisher Scientific | Rn00821446_g1 | |
| Sequence-based reagent | FAM labeled *Kdm6b* (Taqman probe) | ThermoFisher Scientific | Rn01471506_m1 | |
| Sequence-based reagent | VIC labeled *rat β actin* (Taqman probe) | ThermoFisher Scientific | 4352340E | |

*Continued on next page*

*Continued*

| Reagent type (species) or resource | Designation | Source or reference | Identifiers | Additional information |
|---|---|---|---|---|
| Software, algorithm | Graphpad Prism | | RRID:SCR_002798 | |
| Software, algorithm | Adobe Illustrator | | RRID:SCR_010279 | |
| Software, algorithm | Adobe Photoshop | | RRID:SCR_014199 | |

## Cell lines and in vitro tissue culture studies

Immature primary cortical neurons were isolated from CD1 mice embryos (embryonic day 15 [E15]) as previously described (*Ratan et al., 1994*) by following the protocol approved by IACUC at Weill Cornell Medicine. SH-SY5Y human neuroblastoma cells (purchased from ATCC) were cultured in DMEM/F-12 plus GlutaMAX medium added with 10 % fetal bovine serum (Invitrogen) and 1 % penicillin/streptomycin (Invitrogen). HeLa cells and Hep3B cells (purchased from ATCC) were cultured in EMEM medium added with 10 % fetal bovine serum (Invitrogen) and 1 % penicillin/streptomycin (Invitrogen). All cell lines were pre-authenticated by ATCC using STR profiling and were reported mycoplasma negative.

Islets were harvested from Sprague–Dawley male rats (~250 g, Envigo, Huntingdon, Cambridgeshire, United Kingdom) anesthetized by an intraperitoneal injection of sodium pentobarbital (35 mg/230 g rat). All procedures were approved by the University of Washington Institutional Animal Care and Use Committee. Islets were prepared and purified as previously described (*Sweet et al., 2004*) and then cultured at 37 °C in RPMI Media 1640 (Gibco, Grand Island, NY) supplemented with 10 % heat-inactivated fetal bovine serum (Atlanta Biologicals, Lawrenceville, GA) for specified times with the adenovirus coding the $H_2O_2$-sensitive dye (HyPer).

## Adenoviral transduction and normoxia / hypoxia exposure

Adenoviral constructs of MnSOD, catalase, GPX1 and respective GFP control were purchased from University of Iowa, Viral Vector Core Facility. Adenoviral construct of ddGPX4 (GPX4 with destabilization domain) and the respective GFP control were obtained from ViraQuest, Inc (North liberty, IA) and adenoviral constructs of PRDX3, GRX1 and respective GFP control were obtained from Vector Biolabs. For GPX4 construct, we took leverage of a novel technique recently developed by Thomas Wandless group (*Iwamoto et al., 2010*) to deliberately regulate the level of expression of a protein of interest. Adenoviral construct of ddGPX4 had an *E. coli* dihydrofolate reductase (ecDHFR) mutant (it was called as degradation domain) fused to its CMV promoter which displays Trimethoprim (TMP) dependent stability. Because of the fusion of degradation domain to the GPX4 promoter, GPX4 also displayed TMP dependent stability. Without TMP, GPX4 was rapidly degraded completely through proteasome but with increase in dose of TMP, GPX4 stability increases. Treatment of ddGPX4 expressing cells with 10 μM TMP for 12 hr stabilized ddGPX4 very well. SH-SY5Y cells, HeLa cells or Hep3B cells were transduced with different adenoviral constructs at 500 MOI (Multiplicity of infection) incubated for 72 h and primary immature cortical neurons (PCNs) were transduced with different adenoviral constructs at 100 MOI (Multiplicity of infection) for 48 h. The maximal expression of these constructs was determined by expressing adenoviral construct of GFP at 500 MOI for 72 hr in SH-SY5Y cells and 100 MOI for 48 hr in PCNs, respectively, on slides and staining them with GFP antibody (Abcam, ab6556). Ten μM TMP was added to ddGPX4 expressing cells and their respective control GFP expressing cells after 60 hr of adenoviral incubation for 12 hr to get stabilized GPX4 expression. Parallel sets without transduction of adenoviral constructs but probed with same GFP antibody were used as respective negative controls in each in-vitro models. Thereafter, one set was kept in normoxia (21 % oxygen) and a parallel set in hypoxia (1 % oxygen) for 4 hr. For the study of changes in endogenous antioxidants, SH-SY5Y cells and primary cortical neurons were exposed to normoxia/hypoxia for a relatively longer time period, 8 hr so that we could visualize compensatory changes in antioxidants under normoxic and hypoxic conditions.

In order to study real time changes in hydrogen peroxide in hypoxia condition, the adenovirus containing the cytosolic $H_2O_2$ sensor, pHyper-cyto vector (FP942, Evrogen, Moscow, Russia) (*Belousov et al., 2006*) was generated by Vector Biolabs (Malvern, PA) as previously described (*Karamanlidis et al., 2013*). The $H_2O_2$ sensor was transduced in intact islets during incubation in RPMI media supplemented with 10 % heat-inactivated fetal bovine serum and the adenoviruses at 100 MOI for 3 days at

37 °C as previously optimized (**Neal et al., 2016**). SH-SY5Y cells and Hep3B cells were also transduced with the $H_2O_2$ sensor at 100 MOI in a similar manner.

## Real-time epifluorescent imaging of intracellular $H_2O_2$

Real-time imaging experiments were carried out while islets were perifused using a commercially available temperature-controlled Bioptechs FCS2, a closed system, parallel plate flow chamber (Butler, PA) as previously described (**Neal et al., 2016**). After the islets were loaded into the perifusion chambers, the chamber was sealed and mounted onto the stage of a Nikon Eclipse TE-200 inverted microscope. KRB was pumped through the perifusion chamber at flow rates of 120 µl/min using a Masterflex L/S peristaltic pump (Cole-Parmer, Vernon Hills, IL). Use of an artificial gas exchanger positioned on the inflow side of the perifusion chamber enabled rapid changes in the concentrations of dissolved oxygen by switching the source tank of gas between tanks containing 21% and 1% oxygen (balance 5 % carbon dioxide and nitrogen) (**Sweet et al., 2002**). The HyPer signal was generated by dual fluorescence excitation via a xenon arc lamp (Lambda LS-1620, Sutter Instrument Company, Novato, CA) through either a 405/30 nm or a 480/40 nm bandpass filter and detected at 510 nm through a longpass dichroic mirror with a cutoff below 500 nm. The images were taken using a digital camera (Photometrics Cool Snap HQ2 CCD camera, Tucson, AZ) through a 40 X Super Fluor Nikon objective (DIC H/N2). Data were expressed ratiometrically, where the excitation intensities at 480 nm were divided by those obtained during excitation at 405 nm. Similar procedure was used for real time monitoring of $H_2O_2$ in SH-SY5Y cells and Hep3B cells. In order to further confirm the specificity of HyPer signals, we treated Hep3B cells with the bacterial Streptolysin-O (which creates pores in the cell membranes) to selectively permeabilize the plasma membrane followed by their exposure to increasing concentrations of exogenous $H_2O_2$. Real time measurement of changes in the HyPer signals in response to exogenous addition of $H_2O_2$ confirmed the specificity of HyPer signal with regard to $H_2O_2$.

## Real-time epifluorescent imaging of intracellular NAD(P)H

NAD(P)H autofluorescence was measured similarly to $H_2O_2$, except there was no need for loading with dye, and the excitation and emission wavelengths were 360 and 460 nm, respectively (as previously described [**Gilbert et al., 2008**]). To calibrate the relative fluorescence units (RFU), at the end of experiments, the steady state RFU in the presence of potassium cyanide (KCN) and, subsequently, FCCP was measured. The normalized fluorescence of NAD(P)H was then calculated as follows,

$$\%\text{Reduced NAD(P)H} = 100 * (\text{RFU}_{test} - \text{RFU}_{FCCP})/(\text{RFU}_{KCN} - \text{RFU}_{FCCP}) \tag{1}$$

where $\text{RFU}_{FCCP}$ and $\text{RFU}_{KCN}$ equals the average of the final 10 time points during which each agent was present.

## Real-time measurement of insulin secretion rate

Outflow fractions from the flow system containing islets were collected in a fraction collection for subsequent measurement of insulin (described previously **Sweet et al., 2004**; **Sweet et al., 2002**). Insulin secretion rate was calculated as the flow rate (80 µl/min) times the insulin concentration in the outflow Fractions, over the number of islets in the chamber (typically 50 **Sweet and Gilbert, 2006**). Insulin was measured by radioimmunoassay (RI-13K, EMD Millipore, Darmstadt, Germany) as per the manufacturer instructions.

## ODD-luciferase activity assay

The ODD-luciferase construct with pcDNA3.1 plasmid vector was constructed as previously described (**Safran et al., 2006**). The proline p402 and p564 present within the oxygen degradation domain (ODD) of HIF1α, when hydroxylated by HIF-PHDs, allow its binding to the VHL protein that target it for proteasomal degradation. In this way, the stabilization of ODD can be used as a marker of HIF1α stability (**Safran et al., 2006**; **Smirnova et al., 2010**). Because of the luciferase tagged with ODD, the increase in ODD stability leads to a proportional increase in the luciferase activity and this provides a very good way of measuring the HIF1α stability in a quantitative manner with a wide dynamic range. To this end, we used SH-SY5Y cells stably expressing ODD-luciferase. These cells were made by co-transfecting ODD-luciferase plasmid along with a puromycin resistance plasmid in SH-SY5Y cells and stably transfected cells were positively selected in presence of 4 µg/ml of puromycin. Luciferase activity

was measured by luciferase assay kit (promega) using an LMaxII microplate luminometer (molecular Devices). ODD-luciferase activity was normalized to the protein content of each well measured by Bio-Rad DC protein assay kit.

## Gene expression study

Total RNA was prepared from SH-SY5Y cells using the Nucleospin RNA kit (MACHEREY-NAGEL) and following their protocol. Real-time PCRs were performed as a duplex reaction using FAM labeled *Enolase2* (Human - Hs00157360_m1), *Bnip3* (Human - Hs00969291_m1), and *Hif1a* (Human - Hs00153153_m1) gene expression assays (Thermo Fisher Scientific) and VIC-labeled *human β actin* endogenous control probe (Human - 4326315E) or *RNA28S5* (Human - Hs03654441_s1) (Thermo Fisher Scientific) so that amplified mRNA can be normalized to *β actin* or *RNA28S5*. These experiments were performed using a 7500 Real-time PCR system (Applied Biosystems) using standard PCR protocol and amplification conditions.

In order to measure gene expression in pancreatic islets exposed to normoxia or hypoxia, islets were first placed in incubators containing either 21 % or 1 % oxygen for 2 hr. At the end of this time, islets were lysed and total RNA was purified using RNeasy Mini Kit (Qiagen, Hilden Germany). *Bnip3*, *Kdm6b* and *rat Actin B* mRNA were measured by quantitative PCR using FAM-labeled *Bnip3* (Rat - Rn00821446_g1), *Kdm6b* (Rat - Rn01471506_m1) gene expression assays and VIC-labeled *rat β actin* endogenous control probe (Rat - 4352340E), all purchased from Thermo Fisher Scientific. These experiments were performed on Mx3005P Multiplex QPCR System (Stratagene, La Jolla, CA) with samples loaded in triplicate using ~ 100 ng of total RNA.

## Enzyme activity assay

For GPX1, catalase, and MnSOD activity assay, cells in each sample expressing respective adenoviral constructs were collected, lysed and used for respective enzyme activity assay following the protocols of GPX1, catalase, and SOD assay kit from Biovision. Total protein was measured using the Bio-Rad DC protein assay kit. The enzyme activity was normalized to the protein concentration for each sample.

## Cell viability assay

In order to test the functional activity of GPX4, immature primary cortical neurons (E15) were isolated from mice embryos and plated at $10^6$ cells/ml in 96-well plate. The next day, cells were transduced with GPX4 adenoviral constructs at 100 MOI. After 24 hr incubation, cells were treated with glutamate analog, homocysteic acid (HCA) (5 mM) which inhibits Xc- transporter, thereby inhibiting cysteine uptake which leads to glutathione depletion and increase in intracellular oxidative stress. Cells were also treated with 10 μM TMP at the same time to stabilize GPX4 protein. The next day, cell viability was assessed by the MTT assay (Promega) to understand whether ddGPX4 is functionally active to show its protective effect by decreasing oxidants under oxidative stress.

## Cell death assay

SH-SY5Y ODD-luc cells overexpressing different adenoviral constructs of antioxidants were treated with increasing concentrations of H2O2 (1mM – 10mM) and then were exposed to normoxia or hypoxia in parallel for 4 hr and then plates were taken out and media was removed gently and 1xPBS containing 10 μM Sytox blue dye (Nucleic acid stain that easily penetrates cells with compromised membranes and binds to DNA) was added in each well and incubated at room temperature for 15 min and then fluorescence was measured using fluorescence plate reader with excitation/emission maxima at 440/480 nm.

## ROS measurement through DCF flow cytometry

To measure changes in ROS levels, molecular probe DCFDA (2', 7' – dichlorofluorescin diacetate) was used. After exposing parallel sets of SH-SY5Y ODD-Luc cells to either normoxia or hypoxia for 4 hr, media was removed, cells were washed with 1xPBS once, and then 1xPBS with 20 μM DCFDA was added in each well. Additional parallel sets of cells from both conditions were also treated with 5 mM $H_2O_2$ at the same time and were used as positive controls. Plates from both normoxia and hypoxia were then incubated at room temperature for 30 min in normoxic condition. DCFDA diffuses through the cell membrane and is deacetylated by intracellular esterases to a non-fluorescent form which is

later oxidized by ROS into 2′, 7′ – dichlorofluorescein (DCF) which is a highly fluorescent compound. After incubation, fluorescence was measured through flow cytometry at the wavelength of excitation, 485 nm and emission, 535 nm, respectively. Production of ROS was measured as mean fluorescence index multiplied by respective cell counts and was expressed as fold change with respect to control.

## Immunoblotting

Protein extracts were prepared using 1 % triton buffer containing protease inhibitor cocktail, MG132, PMSF, and DTT and were separated by SDS-PAGE, transferred onto nitrocellulose membrane and probed with antibodies against GFP (Cell Signaling Technology; 2555), MnSOD (Sigma-Aldrich; HPA001814), catalase (Sigma-Aldrich; C0979), GPX1 (Cell Signaling Technology; 3,286 S and Novus Biologicals; NBP1-33620), GPX4 (LSBio; LS B1596), PRDX3 (Novus Biologicals; NBP2-19777), Luciferase (Santa Cruz Biotechnology; sc-74548), HIF1α (Novus Biologicals; NB100-479 and NB100-105), Citrate synthase (Cell Signaling Technology; 14,309 S) antibodies were used for immunoblotting. Nuclear-cytoplasmic fractionation was done using NE-PER Nuclear and Cytoplasmic Extraction kit (ThermoFisher Scientific, Catalog number: 78835) and nuclear fraction was used for the immunoblotting of HIF1α using the monoclonal antibody (NB100-105) while the whole cell extract was used for the immunoblotting of HIF1α using polyclonal antibody (NB100-479) and all other immunoblots.

## Quantification and statistical analysis

All experiments were performed as at least three independent sets and data were displayed as means ± standard deviation (SD). Statistical significances were assessed in GraphPad Prism using either Student's t tests to compare values between two specific groups or one-way ANOVA followed by Dunnett's post-hoc test/Tukey's Post-hoc test to compare the values of more than two groups or two-way ANOVA followed by Bonferroni's post-hoc test to compare the values of two groups under two different conditions at a given time. Statistical details for each figure can be found in their respective figure legends. The p value of 0.05 or less was considered statistically significant in all statistical analyses.

## Online supplemental material

*Figure 2—figure supplement 1* shows that hypoxic exposure of SH-SY5Y cells for different time periods such as 2 , 4 , and 8 hr do not induce homeostatic increase in antioxidant enzymes. *Figure 2—figure supplement 2* provides additional evidence in SH-SY5Y cells or Hep3B cells that hypoxia does not increase peroxide levels. *Figure 3—figure supplement 1* shows a validation of the degree of expression of transduced transgenes encoded within adenoviral vectors in SH-SY5Y cell and primary cortical neurons (PCNs) using adenoviral particles encoding GFP. *Figure 3—figure supplement 2* shows the validation of the enzyme activities of various antioxidants in normoxia and hypoxia. *Figure 3—figure supplement 3* shows the validation of the functional activity of antioxidants to decrease reactive oxygen species. *Figure 3—figure supplement 4* provides the additional evidence in primary neurons that the stabilization of HIF1α is not oxidant-initiated in hypoxia. *Figure 3—figure supplement 5* provides additional evidence in HELA cells that the stabilization of HIF1α is not oxidant-initiated in hypoxia. *Figure 3—figure supplement 6* provides another additional evidence in Hep3B cells that the stabilization of HIF1α is not oxidant-initiated in hypoxia. *Figure 5—figure supplement 1* shows the validation of the functional activity of GPX4 to reduce reactive lipid species.

## Acknowledgements

This work was supported by the National Institute of Health (Grant P01 AG14930-15A1, Project one to RRR), by Dr. Miriam and Sheldon G Adelson Medical Research Foundation grant to RRR, by a Goldsmith Fellowship to Amit Kumar for transition to independence, by Diabetes Research Center Cell Function Analysis Core (P30 DK17047; University of Washington) to Ian Sweet. We also want to thank Sunghee Cho and Jiwon Yang for their help in data acquisition through flow cytometry. We acknowledge critical comments and suggestions from Drs. Ratcliffe, Schofield, Semenza, Silva and Ciechanover. The authors declare no competing financial interests.

## Additional information

### Funding

| Funder | Grant reference number | Author |
|---|---|---|
| Dr. Miriam & Sheldon G. Adelson Medical Research Foundation | | Rajiv R Ratan |
| National Institutes of Health | P01 AG14930-15A1 | Rajiv R Ratan |
| Goldsmith Fellowship | | Amit Kumar |

The funders had no role in study design, data collection and interpretation, or the decision to submit the work for publication.

### Author contributions

Amit Kumar, Conceptualization, Data curation, Formal analysis, Investigation, Methodology, Validation, Visualization, Writing - original draft, Writing - review and editing; Manisha Vaish, Conceptualization, Investigation, Methodology, Writing - review and editing; Saravanan S Karuppagounder, Investigation, Methodology; Irina Gazaryan, Anatoly A Starkov, Elizabeth T Anderson, Sheng Zhang, John T Pinto, Austin M Rountree, Wang Wang, Ian R Sweet, Methodology; John W Cave, Data curation; Rajiv R Ratan, Conceptualization, Funding acquisition, Project administration, Resources, Writing - original draft, Writing - review and editing

### Author ORCIDs

Amit Kumar http://orcid.org/0000-0001-5017-9887
Sheng Zhang http://orcid.org/0000-0001-8206-1007
Ian R Sweet http://orcid.org/0000-0002-7565-1663
Rajiv R Ratan http://orcid.org/0000-0002-9081-2701

### Ethics

All procedures involving animals were approved by the Institutional Animal Care and Use Committee of the Weill Cornell Medical College (Animal protocol number: 2013-0121) and the University of Washington Institutional Animal Care and Use Committee (Animal protocol number: 4091-01) and were in accordance with the guidelines established by the National Institutes of Health (NIH) and ARRIVE (Animal Research Reporting of In Vivo Experiments).

### Decision letter and Author response

Decision letter https://doi.org/10.7554/eLife.72873.sa1
Author response https://doi.org/10.7554/eLife.72873.sa2

## Additional files

### Supplementary files

• Transparent reporting form

### Data availability

All data generated or analysed during this study are included in the manuscript and supporting file.

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
