## [Decision Letter]

**Acceptance summary:**

All reviewers agreed that the manuscript addresses an important question that is of interest and uses fresh approaches and new methodologies to make a significant contribution to resolve this controversial issue of hypoxic signaling. The reviewers also commend the authors for revising their manuscript with additional data and changes in the manuscript text in response to their comments. The reviewers agree that after the revision there are no further concerns.

**Decision letter after peer review:**

[Editors’ note: the authors submitted for reconsideration following the decision after peer review. What follows is the decision letter after the first round of review.]

Thank you for submitting your work entitled "Oxidants are dispensable for HIF1α stability in hypoxia" for consideration by *eLife*. Your article has been reviewed by 3 peer reviewers, including Thilo Hagen as the Reviewing Editor and Reviewer #2, and the evaluation has been overseen by a Senior Editor. The following individual involved in review of your submission has agreed to reveal their identity: Paul S Brookes (Reviewer #3).

Our decision has been reached after consultation between the reviewers. Based on these discussions and the individual reviews below, we regret to inform you that your work will not be considered for publication in *eLife* in its current form.

All reviewers agreed that the manuscript addresses an important question that is of interest. The reviewers acknowledge that the study is generally very thorough and uses fresh approaches and new methodologies to make a significant contribution to resolve this controversial issue of hypoxic signaling. However, there are a number of general criticisms, including a tendency to over-interpret the results and over-state conclusions. As pointed out in the specific comments, there are various instances where the presentation and interpretation of the data is inappropriate and these need to be addressed. The reviewers also made a number of suggestions for additional experiments to support the conclusions of the study or extend the study further. If the authors decide to resubmit their manuscript to *eLife*, all reviewer comments should be seriously considered. It is necessary that the authors respond to each of the reviewers comments in detail.

*Reviewer #1:*

It has long been known that mitochondria are involved in regulating the cellular response to hypoxia, either through increased O2 consumption, through the generation of reactive oxygen species (ROS), or other mechanisms. But there has been controversy over the requirement of ROS in contributing to hypoxia inducible factor -1alpha (HIF-1alpha) in hypoxia, partly due to methodological differences, and different cell systems used from group to group. Previous studies have used antioxidant agents to assess HIF-1alpha stability in hypoxia, and have shown differing effects. This is an interesting area of research, but there are fundamental and major concerns with the approaches taken.

Figure 1. The authors show expression of antioxidant proteins by western and measurement of peroxide levels. 21%O2 (atmospheric) and 1-2% O2 are used.

What is needed are time courses and different O2 levels, especially physiological levels of O2 (5-10%). Also, proteomic analyses would be far more 'state-of-the-art' than a simple western, especially given that no difference are shown on the westerns.

Figure 2. The authors use DFO, and measure activity of an exogenous ODD-luc reporter (as an indirect measurement of HIF). Although DFO, an iron chelator will stabilize HIF1a in cells, but it is highly toxic and non-specific. There are many other 'targeted' PHD inhibitors available (see Nat Rev Nephrol, 2016;12(3):157-68) and should be used here. Activity of exogenously overexpression of antioxidant enzymes by adenovirus shown -is this normoxia only? What activity in about hypoxia?

Figure 3. SH-SY5Y cells stably expressing ODD-luciferase were transduced with adenoviruses encoding distinct antioxidant enzymes for 72h. Catalase (CAT) increases and GPX1 decreased ODD-luc promoter activity in hypoxia. While the ODD-HIF-1a protein levels tally with this finding (3B), the endogenous HIF-1a does not -increased with CAT but does not decrease with GPX1. These data suggest the ODD-luc does not function like the endogenous protein. Can the authors explain this discrepancy?

Figure 4. Authors only show ODD-luc data, but they should show half-life of the endogenous HIF-1a protein, also HIF2a should be shown.

Figure 5. It is not clear how FIH relates to these data, as suggested in the results text (line 301). To properly test the role of FIH, the authors should use FIH KO cells.

Figure 6. The authors show no effect of exogenously expressed GPX4 on HIF1a. The authors should assess HIF2a (see Zho et al. Nat Comm 2019, 10 (1617)).

*Reviewer #2:*

In their study, Kumar et al. characterized the role of mitochondria in hypoxic signaling. This is an important topic, as both mitochondrial function and hypoxic signaling are frequently altered in various pathological conditions. For instance, tumors are commonly hypoxic and activation of the cellular hypoxic response is often critical for tumor cell survival or even promotion of tumorigenesis. At the same time, as a result of induction of the Warburg effect, mitochondrial function is frequently downregulated in cancer cells. In ischemic diseases, the induction of the cellular hypoxic response has an important protective role, while mitochondria derived ROS, especially during reperfusion, are an important contributor to cell and tissue injury. Hence, understanding the link between the cellular hypoxic response and mitochondrial (dys)function is of great interest.

The mechanism through which mitochondria regulate the hypoxic response has been a long standing controversy. On the one hand, it has been proposed that hypoxia causes an increase in mitochondria derived ROS, which promotes the induction of the HIF1alpha transcription factor. On the other hand, mitochondrial activity regulates the intracellular oxygen concentration, which may have implications of hypoxic signaling and other cellular oxygen dependent processes (e.g. histone demethylation and ER protein folding).

What is special about the current study is how rigorous and in depth the authors addressed the question of how mitochondrial function affects the hypoxic response, specifically whether mitochondria-derived ROS are involved. The authors used relevant models and included extensive controls. Examples include the use of non-transformed primary cells as well as rat pancreatic islet cells, which display physiological mitochondria-dependent H2O2 increases in response to elevated glucose. The authors took into account changes in mitochondrial mass when assessing changes in expression levels, used positive controls to validate assays, validated of transfection efficiencies and functional expression levels, and confirmed their results in multiple cell lines and using multiple assays. As such, the study provides a very in depth analysis, which allows to draw firm conclusions.

Furthermore, throughout the study, the authors produced high quality data and went out of their way to show that the experimental settings worked and rule out alternative explanations. In fact, I have rarely reviewed a study that us so well planned, conducted and presented. In summary, the significance of the manuscript does not lie in the novelty, but in the fact that the study addresses an important problem and provides high quality data, which makes the study very conclusive.

I have the following specific comments:

1. My major concern is that the authors did not confirm that mitochondria indeed regulate HIF1α stability in the used cellular system.

2. Results (first paragraph, line 138):

The following sentence implies causality that has not really been shown by the authors.

"… since hypoxia induces HIF1α-dependent mitophagy (Aminova, Siddiq, & Ratan, 2008; Zhang et al., 2008), mitochondrial mass is decreased with increasing duration of hypoxia and includes decreases in mitochondrial DNA and proteins."

3. line 187: Should this say "increases in NADPH levels"?:

"Increasing the glucose concentration from 3 mM to 20 mM elicited the expected increases in peroxide, decreases in NADPH levels and insulin secretion rates under normoxia."

*Reviewer #3:*

In this work, the authors attempt to address a long-standing controversy in the cell signaling literature, namely the role of reactive oxygen species (ROS) in hypoxic cell signaling. Previous claims that ROS are required for the activation of hypoxia inducible factor (HIF) have not been reconciled with the requirement of molecular oxygen as a substrate for ROS generation (i.e. how can there be more ROS in hypoxia when there is less substrate to make it?) A further critique of previous work is that the genetic manipulations applied to alter ROS (namely, knocking out components of the respiratory chain) would also be expected to alter oxygen consumption and thus intra-cellular oxygen gradients. Herein, contemporary approaches are used, including manipulation of several enzymes that degrade ROS, to show that ROS are not required for HIF activation.

In general the work is technically sound, using multiple different cell systems, and often several overlapping methods to achieve the same ends. Much of the reasoning used in drawing conclusions is iterative, and while this is mostly sound, it is somewhat stretched in places. For example, a key observation is that antioxidant enzymes (catalase, GPXs, MnSOD) are not induced by hypoxia, and this is used to argue that ROS are not increased – because if ROS were up then antioxidants should be up too. There are several alternative explanations: (i) Maybe they didn't choose the right time point. (ii) Maybe they didn't look at the right antioxidants. (iii) Maybe antioxidants didn't change because the ROS signal was too small. In general, I feel that the paper errs toward drawing one conclusion without adequate consideration of alternative explanations.

The use of the HyPer probe for H2O2 is a great approach, although one must question the use of the original HyPer from 2006, with its pH sensitivity problems (the authors do address this experimentally, although again the approach is somewhat dated). Newer more sensitive generations of this probe are available (HyPer3 since 2013), so the use of the older probe cannot be termed "state-of-the-art".

The results in Figure S1C are very puzzling. It is widely reported in the literature that cyanide (KCN) does not lead to an increase in mitochondrial ROS. This is because when cytochrome c oxidation is prevented, feedback inhibition occurs into the C1 and Fe-S subunits of complex III. If the Qo site of complex III cannot give up its first electron to the upper pathway, then the semiquinone radical is not generated and so ROS from this site is suppressed. This was the strategy used by Chandel and others to prove the role for the Qo site as the source of ROS – knocking out the Fe-S protein prevents ROS generation. Here, the HyPer signal increases with KCN. A simple (but unfair) interpretation would be that the probe does not measure ROS. But, even if it does, the effect of KCN is still puzzling when compared to that of hypoxia – both of these interventions inhibit complex IV and so should have the same effect, but herein only KCN impacts the probe signal. A more likely explanation here is that KCN may have off target effects in cells – it is known to inhibit peroxidases and catalase, and so likely the increase in probe signal seen with KCN is due to prevention of ROS scavenging by such enzymes.

A key concern is raised with Figure S3A and S3B, in which the authors used the non-specific redox probe DCF to measure oxidants. Notably, they saw no increase in DCF signal in hypoxia. This experiment therefore fails to replicate the fundamental findings of the Chandel/Schumacker lab that started this whole field (namely hypoxia gives rise to an increase in DCF signal). While obtaining different results using a different probe (HyPer) is reasonable and even to be expected, not being able to repeat the fundamental result from the other lab (and many other labs) is somewhat problematic.

Furthermore, the last set of bars on the right in these experiments (Figure S3A/B) clearly show that while addition of H2O2 led to an increase in DCF signal under normoxic conditions, no increase was seen in hypoxia. This seems like an important difference between the cells. One possible explanation is that hypoxic cells had increased H2O2 scavenging ability.

The results in Figure 4, showing that alterations in the half-life of HIF induced by antioxidants expression was due to impacts on proteasomal degradation, are quite important, and buried deep in the paper. Likewise Figure 5 data supporting this HIF stability finding. I feel more could be made of this part of the story.

Overall, while the results are sound, the interpretation of some key experiments is lacking in places. Collectively, the results do point toward the idea that oxidants do not play a major role in HIF activation during hypoxia, and this will be of interest to the field.

Several suggestions to improve the manuscript.

The western blots provided are not up to contemporary standards for publication. The blots are presented as letter-boxed slices without molecular weight markers. In some cases (e.g. Figure 2A) the blots are washed out so there is no background, thus not providing any features to anchor the bands to the gel. It is requested to show original full-height blots with markers in supplemental information.

Some places the western blots used as confirmatory evidence for the reporter assays are presented as N=1. It would be good to add some quantitation to these blots, assuming they were representative of multiple experimental replicates.

A major place for improvement would be to consider and discuss alternative explanations for the results, as outlined above in the public review.

It is a little misleading to label the signal for the HyPer probe "H2O2" (as is the case with all such probes). What is measured is the steady-state of a redox sensitive disulfide bond, which will be somewhat sensitive to H2O2 flux but also other factors such as the counteracting flux of reductants. It is a little unfair to refer to this as an H2O2 probe, and to DCF as a non-specific oxidant probe. While there is no such thing as a perfect ROS probe, and in general genetically encoded sensors offer a number of advantages, caution still needs to be exercised in interpreting results.

If I were to recommend another experiment (at the discretion of the editors and authors due to covid restrictions), it would be to measure NAD(P)H levels in the cells where antioxidants have been manipulated. Since many of these enzymes use NADPH as a substrate, changing their expression could impact the pools of these nucleotides, which could in turn impact redox sensitive processes. For example in Figure 3, the NADPH dependent GPX1 has an effect, but non-NADPH dependent antioxidants (MnSOD, CAT, PRDX3) do not.

Line 188 – text says NADPH decreases upon glucose ramp, but the signal in the figure (panel 1L) increases at this point.

Line 200 – "in" appears twice.

Line 266 – use of the term "reduce" to mean lower/decrease, should be rethought (same on line 298).

The logic flow of the second paragraph of the paper (lines 70-88) is not strong. It starts out saying that one area where ROS signaling has been investigated is hypoxia. Then some information about the nuts and bolts of HIF signaling, and then something about diseases. Then the concluding line "understanding the role of ROS in HIF mediated adaptation has clinical implications". While yes, studying HIF is clinically relevant, the paragraph does not make a convincing case that studying ROS in the context of HIF signaling is relevant. Nothing in the middle makes a convincing case that any of the diseases associated with HIF are actually related to ROS. Some information on how ROS interplay with the HIF system is necessary, along with some details on how HIF is impacted at the molecular level in these diseases.

---

## [Author Response]

[Editors’ note: the authors resubmitted a revised version of the paper for consideration. What follows is the authors’ response to the first round of review.]

Reviewer #1:It has long been known that mitochondria are involved in regulating the cellular response to hypoxia, either through increased O2 consumption, through the generation of reactive oxygen species (ROS), or other mechanisms. But there has been controversy over the requirement of ROS in contributing to hypoxia inducible factor -1alpha (HIF-1alpha) in hypoxia, partly due to methodological differences, and different cell systems used from group to group. Previous studies have used antioxidant agents to assess HIF-1alpha stability in hypoxia, and have shown differing effects. This is an interesting area of research, but there are fundamental and major concerns with the approaches taken.Figure 1. The authors show expression of antioxidant proteins by western and measurement of peroxide levels. 21%O2 (atmospheric) and 1-2% O2 are used.What is needed are time courses and different O2 levels, especially physiological levels of O2 (5-10%). Also, proteomic analyses would be far more 'state-of-the-art' than a simple western, especially given that no difference are shown on the westerns.

We would like to thank the reviewer for this valuable suggestion. As suggested, we performed a time course to see if there is a temporal change in endogenous antioxidant protein levels under hypoxia (1% O_2_), but despite looking at multiple time points, we didn’t detect any change in endogenous antioxidant protein levels under hypoxia (Figure 2-figure supplement 1).

We did not assess higher oxygen tensions (5-10% O2) because in past studies a number of groups have transitioned from 21 % O2 to 1% O2, as we did, to assess the role of ROS in hypoxic HIF stabilization. Also, the reviewer’s suggestion about performing “proteomic analyses” is an important suggestion but since we have already assessed antioxidant enzymes, which have not only different cellular localizations and also work differently as ROS scavengers (Author response image 1), we believe that the compensatory response to changes in hypoxic ROS levels should be reflected by changes in at-least one of these antioxidant enzymes. Therefore, we depended on quantitative changes reflected by immunoblotting of these antioxidant enzymes rather than doing proteomic analyses which can be difficult to quantitate.

We formally concede that there is still a possibility that we might have missed the right antioxidant to examine. However, that was the main reason that we went ahead with direct assessment of changes in ROS levels using real time imaging in HyPer fluorescence. Finally, we would like to point out that no single set of experiments allowed us to reach the conclusion that HIF stability is independent of ROS or RLS. It was the aggregate of multiple approaches that led to the same conclusion.

**Author response image 1. sa2fig1:** Sub-cellular localization of various antioxidants.

Figure 2. The authors use DFO, and measure activity of an exogenous ODD-luc reporter (as an indirect measurement of HIF). Although DFO, an iron chelator will stabilize HIF1a in cells, but it is highly toxic and non-specific. There are many other 'targeted' PHD inhibitors available (see Nat Rev Nephrol, 2016;12(3):157-68) and should be used here. Activity of exogenously overexpression of antioxidant enzymes by adenovirus shown -is this normoxia only? What activity in about hypoxia?

We apologize for the confusion. We had previously used DFO only to examine if the dose-dependant dynamic changes seen in endogenous HIF1α protein level correlate well with dose-dependent dynamic changes seen in exogenous ODD-Luciferase activity in response to increasing concentrations of DFO. As per the reviewer’s suggestion, now, we have used a selective HIF-PhD inhibitor, adaptaquin, which we had identified in an unbiased screen of an 85,000 Rockefeller library compounds and had characterized in collaboration with Dr. Schofield, University of Oxford as a selective HIFPhD inhibitor (Figure 3; Karuppagounder et al., Sci. Transl. Med., 2016). We again treated SH-SY5Y neuroblast cells with increasing concentrations of adaptaquin for 4 hr and assessed dose-dependent changes in either endogenous HIF1α protein level or exogenous ODD-Luciferase activity or expression of endogenous HIF target genes, *Bnip3* and *Eno2*. We, again, not only found a strong correlation between dose-dependent dynamic changes in endogenous HIF1α protein level with dose-dependent dynamic changes seen in exogenous ODD-Luciferase activity but also between endogenous HIF target genes (*Bnip3* and *Eno2*) and exogenous ODD-Luciferase activity (Figure 1). Importantly, the coefficient of variation for HIF1α protein level (assessed through western blot) between one set to another set was much higher compared to ODD-Luc activity. This provided us more confidence in using ODD-Luc activity as a surrogate assessment for quantitative measurement of changes in endogenous level of HIF1α protein.

The reviewer’s question about the enzyme activity assay was a valid question but a technically challenging one to address as once we take samples out of hypoxic chamber, enzymatic activities of antioxidants may change due redox changes introduced by normoxia. We, therefore, used an alternative approach to address this issue. We treated SHSY- 5Y neuroblast cells with increasing concentrations of H2O2 and then exposed one set to the normoxia and other set to the hypoxia (1% O2) for four hours and thereafter assessed cell death using a nuclear dye Sytox blue. As expected, MnSOD overexpression, which should increase peroxide did not protect cells from H2O2 led cell death. Of note, the pattern of death was similar in both normoxia and hypoxia indicating that hypoxia itself did not affect the enzyme activity of MnSOD. Importantly, catalase and GPX1 overexpression protected these cells both in normoxia and hypoxia indicating that hypoxia itself did not affect the enzyme activities of these overexpressing antioxidant constructs (Figure 3-figure supplement 2).

Figure 3. SH-SY5Y cells stably expressing ODD-luciferase were transduced with adenoviruses encoding distinct antioxidant enzymes for 72h. Catalase (CAT) increases and GPX1 decreased ODD-luc promoter activity in hypoxia. While the ODD-HIF-1a protein levels tally with this finding (3B), the endogenous HIF-1a does not -increased with CAT but does not decrease with GPX1. These data suggest the ODD-luc does not function like the endogenous protein. Can the authors explain this discrepancy?

Thank you for pointing this out. We ran the western blot of endogenous HIF1α again in three sets using another monoclonal HIF1α antibody (Novus Biologicals, Catalog number: NB 100-105) (Author response image 2). We agree that the particular representative blot shown earlier in that Figure 3 was confusing. However, the densitometric analysis of an average of three new sets showed expected decreases in HIF1α prote in levels in response to GPX1 overexpression, which we did not include earlier. Now, we have included the new representative HIF1α western blot in Figure 3. We have now also included densitometric analyses of the western blots from all three sets to avoid this confusion. Due to a high coefficient of variation between different sets of HIF1α western blots, changes in HIF1α protein levels in response to catalase or GPX1 were not statistically significant as compared to that of GFP. However, similar changes reflected in ODD-Luc activity and HIF target genes, *Trib3* and *Eno2,* were statistically significant as compared to that of GFP, which we have now moved from Figure 5 to Figure 3 to show the correlation between changes in ODD and HIF1α protein with the transcriptional activity of HIF1α.

**Author response image 2. sa2fig2:** Relative changes in HIF1α protein level in response to overexpression of MnSOD, catalase or GPX1 under normoxia and hypoxia.

Figure 4. Authors only show ODD-luc data, but they should show half-life of the endogenous HIF-1a protein, also HIF2a should be shown.

Since we have not only found a strong correlation between ODD-Luciferase activity and HIF1α protein level but also with the transcriptional activity of HIF1α (Figure 1), we preferred to depend on the findings from half-life of ODD-Luciferase activity rather than examining the half-life of HIF1α.

Also, the suggestion to examine changes in HIF2α is a great one but since the focus of our study is to understand if ROS regulate hypoxic HIF1α stability or not, we preferred to focus mainly on HIF1α rather than HIF2α in order to avoid confusing readers. We are planning to examine HIF2α in a detailed fashion as a follow up project.

Figure 5. It is not clear how FIH relates to these data, as suggested in the results text (line 301). To properly test the role of FIH, the authors should use FIH KO cells.

We did not pursue the FIH knockout studies as suggested by the reviewer for several reasons. First, in all of our manipulations, we always saw a good correlation between HIF-1alpha levels and transcriptional activity suggesting that protein stability via the HIF PHDs and activity of FIH were highly correlated. Second, since forced expression of enzymes that neutralized ROS or RLS did not inhibit transcriptional activity in the setting of increased HIF-1 stability, these results argue against FIH as a target for ROS or RLS neutralized by GPX4. Since we did not pursue detailed FIH knockout studies, we have now avoided the focus on FIH in the manuscript.

Figure 6. The authors show no effect of exogenously expressed GPX4 on HIF1a. The authors should assess HIF2a (see Zho et al. Nat Comm 2019, 10 (1617)).

Thank you for bringing our attention to the Zho et al. (Nat Comm 2019, 10 (1617)) where authors have found that HIF2α enriches cells with polyunsaturated fatty acids (PUFA) and thereby, sensitizes cells to ferroptosis. Notably, Zho et al. did not shown that Lipids or GPX4 (which neutralizes reactive lipids) regulates hypoxic HIF2α stability. It would be interesting to examine if GPX4 regulates the hypoxic HIF2α stability but as the focus of the current study is to understand the regulation of hypoxic HIF1α stability, we plan to undertake this interesting question in another a follow-up project heavily focused on hypoxic HIF2α stability.

Importantly, we would refer the reviewer to a paper published by our group in 2004 in JBC entitled ,”Prosurvival and prodeath effects of hypoxia-inducible factor-1 α Stabilization in a murine cell line” (Aminova et al. 2005). In that manuscript, we showed that forced expression of a HIF-1alpha VP-16 fusion protein potentiates ferroptosis; conversely reducing HIF-1alpha via shRNA rescued ferroptotic death. Although it was not labeled ferroptosis in 2004, cell death induced by glutamate and potentiated by HIF-1 expression is the result primarily of Xc- transport inhibition, the same target inhibited by erastin, the canonical Xc- transport inhibitor. Of note, ferroptosis does not induce HIF-1 stability on its own (Zaman et al., 1999).

Reviewer #2:In their study, Kumar et al. characterized the role of mitochondria in hypoxic signaling. This is an important topic, as both mitochondrial function and hypoxic signaling are frequently altered in various pathological conditions. For instance, tumors are commonly hypoxic and activation of the cellular hypoxic response is often critical for tumor cell survival or even promotion of tumorigenesis. At the same time, as a result of induction of the Warburg effect, mitochondrial function is frequently downregulated in cancer cells. In ischemic diseases, the induction of the cellular hypoxic response has an important protective role, while mitochondria derived ROS, especially during reperfusion, are an important contributor to cell and tissue injury. Hence, understanding the link between the cellular hypoxic response and mitochondrial (dys)function is of great interest.The mechanism through which mitochondria regulate the hypoxic response has been a long standing controversy. On the one hand, it has been proposed that hypoxia causes an increase in mitochondria derived ROS, which promotes the induction of the HIF1alpha transcription factor. On the other hand, mitochondrial activity regulates the intracellular oxygen concentration, which may have implications of hypoxic signaling and other cellular oxygen dependent processes (e.g. histone demethylation and ER protein folding).What is special about the current study is how rigorous and in depth the authors addressed the question of how mitochondrial function affects the hypoxic response, specifically whether mitochondria-derived ROS are involved. The authors used relevant models and included extensive controls. Examples include the use of non-transformed primary cells as well as rat pancreatic islet cells, which display physiological mitochondria-dependent H2O2 increases in response to elevated glucose. The authors took into account changes in mitochondrial mass when assessing changes in expression levels, used positive controls to validate assays, validated of transfection efficiencies and functional expression levels, and confirmed their results in multiple cell lines and using multiple assays. As such, the study provides a very in depth analysis, which allows to draw firm conclusions.Furthermore, throughout the study, the authors produced high quality data and went out of their way to show that the experimental settings worked and rule out alternative explanations. In fact, I have rarely reviewed a study that us so well planned, conducted and presented. In summary, the significance of the manuscript does not lie in the novelty, but in the fact that the study addresses an important problem and provides high quality data, which makes the study very conclusive.I have the following specific comments:1. My major concern is that the authors did not confirm that mitochondria indeed regulate HIF1α stability in the used cellular system.

We would like to thank the reviewer for this important question. As suggested, we tested the involvement of mitochondria in hypoxic HIF1α stability in SH-SY5Y ODD-Luc cells using mitochondrial electron transport chain complex inhibitors such as rotenone (complex I inhibitor), myxothiazol (complex III inhibitor), antimycin A (complex III inhibitor), and sodium azide (complex IV inhibitor). We were able to see that these inhibitors blocked the hypoxic HIF1α stability in a dose dependent manner indicating that mitochondria are involved in mediating the hypoxic HIF1α stability in SH-SY5Y neuroblast cell line (Figure 1).

2. Results (first paragraph, line 138):The following sentence implies causality that has not really been shown by the authors."… since hypoxia induces HIF1α-dependent mitophagy (Aminova, Siddiq, & Ratan, 2008; Zhang et al., 2008), mitochondrial mass is decreased with increasing duration of hypoxia and includes decreases in mitochondrial DNA and proteins."

Thank you for the question. We apologize for the confusion. The goal behind citing the above prior findings from our group and other groups was to emphasize that we have chosen earlier time points following hypoxia rather than later time points in our studies to avoid the mitophagy process as it will also likely change ROS production and/oxygen consumption. The additional changes in mitochondrial proteins due to mitophagy will complicate the interpretation of our results. Moreover, the goal behind mentioning above sentence was not to prove causality of HIF1α in HIF1α-dependent mitophagy, which has been already shown in other studies. Rather, we preferred not to pursue issues related to HIF1α-dependent mitophagy directly but rather added experimental data related to its occurrence.

3. line 187: Should this say "increases in NADPH levels"?:"Increasing the glucose concentration from 3 mM to 20 mM elicited the expected increases in peroxide, decreases in NADPH levels and insulin secretion rates under normoxia."

Thank you for pointing this out. We apologize for this typing mistake. We have now corrected the sentence in the manuscript as “Increasing the glucose concentration from 3 mM to 20 mM elicited the expected increases in peroxide, NADPH levels and insulin secretion rates under normoxia."

Reviewer #3:In this work, the authors attempt to address a long-standing controversy in the cell signaling literature, namely the role of reactive oxygen species (ROS) in hypoxic cell signaling. Previous claims that ROS are required for the activation of hypoxia inducible factor (HIF) have not been reconciled with the requirement of molecular oxygen as a substrate for ROS generation (i.e. how can there be more ROS in hypoxia when there is less substrate to make it?) A further critique of previous work is that the genetic manipulations applied to alter ROS (namely, knocking out components of the respiratory chain) would also be expected to alter oxygen consumption and thus intra-cellular oxygen gradients. Herein, contemporary approaches are used, including manipulation of several enzymes that degrade ROS, to show that ROS are not required for HIF activation.In general the work is technically sound, using multiple different cell systems, and often several overlapping methods to achieve the same ends. Much of the reasoning used in drawing conclusions is iterative, and while this is mostly sound, it is somewhat stretched in places. For example, a key observation is that antioxidant enzymes (catalase, GPXs, MnSOD) are not induced by hypoxia, and this is used to argue that ROS are not increased – because if ROS were up then antioxidants should be up too. There are several alternative explanations: (i) Maybe they didn't choose the right time point. (ii) Maybe they didn't look at the right antioxidants. (iii) Maybe antioxidants didn't change because the ROS signal was too small. In general, I feel that the paper errs toward drawing one conclusion without adequate consideration of alternative explanations.

Thank you for providing this important logical feedback. Now, we have corrected the interpretation of our results by including alternate possibilities in the manuscript as suggested. We have also performed a more extensive time course of antioxidant enzyme activities. We want to emphasize that no single set of experiments in our manuscript can drive the conclusion that ROS are likely not important. Rather, the combination of a lack of change in antioxidant enzymes; the inability to detect a change in ROS using a selective, ratiometric reporter; and the lack of consistent effect of antioxidant enzyme manipulation on HIF stability together argue that ROS are not likely important for HIF stability

The use of the HyPer probe for H2O2 is a great approach, although one must question the use of the original HyPer from 2006, with its pH sensitivity problems (the authors do address this experimentally, although again the approach is somewhat dated). Newer more sensitive generations of this probe are available (HyPer3 since 2013), so the use of the older probe cannot be termed "state-of-the-art".

We concede that this newer probe is reported to be more sensitive and also have quicker dynamics (although there is no mention in the 2013 paper of not having the same pH dependency). We have now removed the term “state-of-the-art” from the main text as suggested. We utilized the current version of Hyper because it was extensively validated in a prior manuscript by one of our groups (Neal et al. 2016).

The results in Figure S1C are very puzzling. It is widely reported in the literature that cyanide (KCN) does not lead to an increase in mitochondrial ROS. This is because when cytochrome c oxidation is prevented, feedback inhibition occurs into the C1 and Fe-S subunits of complex III. If the Qo site of complex III cannot give up its first electron to the upper pathway, then the semiquinone radical is not generated and so ROS from this site is suppressed. This was the strategy used by Chandel and others to prove the role for the Qo site as the source of ROS – knocking out the Fe-S protein prevents ROS generation. Here, the HyPer signal increases with KCN. A simple (but unfair) interpretation would be that the probe does not measure ROS. But, even if it does, the effect of KCN is still puzzling when compared to that of hypoxia – both of these interventions inhibit complex IV and so should have the same effect, but herein only KCN impacts the probe signal. A more likely explanation here is that KCN may have off target effects in cells – it is known to inhibit peroxidases and catalase, and so likely the increase in probe signal seen with KCN is due to prevention of ROS scavenging by such enzymes.

Thank you for this logical comment. This explanation is plausible, and our data cannot distinguish between an increase in H2O2 being mediated by an increase production rate, or a decrease in rate of destruction. It is likely that (1) KCN inhibits peroxidases and catalases and leads to an increase in ROS or (2) hypoxia decreases the rate of H2O2 formation due to a lack of O2, and KCN increases the rate of H2O2 due to increased transfer of electron from highly reduced ETC proteins to plentiful O2 in normoxia. This scenario is supported by the increase in NAD(P)H and our past studies on hypoxia and reductive state of cytochromes [Sweet IR et al., *Diabetes Tech. Ther.*, 2002. (Figure 7); Sweet IR et al., *Diabetes* 2004 (Figure 3)]. Since we cannot rule out the scenario 1 and we have measured cytosolic H2O2 using cyto-HyPer and not mitochondrial H2O2, we preferred to remove the HyPer data in response to KCN treatment from figures S2C and S2G. However, since it is a positive control, the mechanism is not as important as the fact that it increases ROS and that this can be detected by the reporter.

A key concern is raised with Figure S3A and S3B, in which the authors used the non-specific redox probe DCF to measure oxidants. Notably, they saw no increase in DCF signal in hypoxia. This experiment therefore fails to replicate the fundamental findings of the Chandel/Schumacker lab that started this whole field (namely hypoxia gives rise to an increase in DCF signal). While obtaining different results using a different probe (HyPer) is reasonable and even to be expected, not being able to repeat the fundamental result from the other lab (and many other labs) is somewhat problematic.Furthermore, the last set of bars on the right in these experiments (Figure S3A/B) clearly show that while addition of H2O2 led to an increase in DCF signal under normoxic conditions, no increase was seen in hypoxia. This seems like an important difference between the cells. One possible explanation is that hypoxic cells had increased H2O2 scavenging ability.

Thank you for pointing this out. A possible reason behind not seeing an increase in DCF signal under hypoxia as compared to normoxia is the way we have assessed the DCF signal. The way we have used DCF to assess oxidants was different from the method used by Dr. Chandel and others. Others had added DCF in the media before exposing cells to Normoxia/hypoxia and had kept it onboard during the exposure whereas we first exposed cells to Normoxia/hypoxia for 4 hours and then removed the media and washed cells once with 1X PBS and then added 100 microliters of 1X PBS containing 20μM DCF in each well. The reason we have done this way is that when we add DCF before the exposure and keep it onboard during the exposure, we are not sure if the increase in DCF is because of an increase in oxidant signaling or because of an increased accumulation of DCF in hypoxic cells as compared to normoxic cells. However, when we add DCF after completing the exposure of cells to normoxia/hypoxia and incubate these normoxic/hypoxic cells in the same environment for 30 min, we minimize the possibility of differential accumulation of DCF by these cells. We used simultaneous H2O2 treatment for 30 min as a positive control. Thereafter, we measured DCF through flow cytometry to get population measurement of change in mean fluorescence intensity. In our findings, we did not find an increase in DCF signaling in cells which were exposed to hypoxia for four hours as compared to those from normoxia. Additionally, as pointed out by the reviewer, we did not find an increase in DCF signaling with H2O2 in cells, which were exposed to hypoxia. This could be because of either decrease in ROS level during hypoxia (as can be seen with control GFP hypoxia compared with control GFP normoxia), which likely increased with H2O2 treatment to the level of GFP control under normoxia but didn’t go further up or because of increased scavenging capacity of SH-SY5Y cells under hypoxia as mentioned by the distinguished reviewer. We have now discussed these possibilities in the Results section of the manuscript to make results clearer.

Importantly, we did find an increase in DCF signal under hypoxia as Dr. Chandel and other groups have reported before by following their method i.e. by adding DCF before exposing cells to normoxia/hypoxia and keeping it onboard during the exposure but as you know DCF integrates oxidation so it is difficult to establish whether changes in signal are due to changes in excretion, cell geometry, or dye concentrations. That is why, we had chosen to add DCF after exposing cells to normoxia/hypoxia for a brief period of time and use flow cytometry.

We understand that there are limitations associated with using DCF for assessing changes in ROS level, hence our use of Hyper. But based on the above mentioned rationale, we preferred to confirm functional activities of antioxidants using DCF using a method that would diminish effects unrelated to ROS sensing. We know that this is the reason behind not seeing an increase in DCF signaling in hypoxia. We tried to be as careful as possible to avoid the non-specific measurements associated with DCF.

The results in Figure 4, showing that alterations in the half-life of HIF induced by antioxidants expression was due to impacts on proteasomal degradation, are quite important, and buried deep in the paper. Likewise Figure 5 data supporting this HIF stability finding. I feel more could be made of this part of the story.

Thank you for making this important comment. We have now tried to make the story more coherent and clear based on reviewer’s important comments. Now, we have merged the gene expression data of HIF1α target genes, *Bnip3* and *Eno2* from Figure 5 to Figure 3 to show it next to ODD-Luciferase activity and HIF1α immunoblots in order to consolidate these important findings together.

Overall, while the results are sound, the interpretation of some key experiments is lacking in places. Collectively, the results do point toward the idea that oxidants do not play a major role in HIF activation during hypoxia, and this will be of interest to the field.

Thank you again for this important comment. We have now revised and clarified the interpretation of our experiments in a number of places. We hope this will convince the reviewer that our results are sound.

Several suggestions to improve the manuscript…The western blots provided are not up to contemporary standards for publication. The blots are presented as letter-boxed slices without molecular weight markers. In some cases (e.g. Figure 2A) the blots are washed out so there is no background, thus not providing any features to anchor the bands to the gel. It is requested to show original full-height blots with markers in supplemental information.

Thank you for this important comment. We have also replaced the washed out blots with original cropped blots without any change in intensity of the blots in all figures. We have now also included the original full-height blots of all representative blots with molecular weight markers in the supplementary information as suggested.

Some places the western blots used as confirmatory evidence for the reporter assays are presented as N=1. It would be good to add some quantitation to these blots, assuming they were representative of multiple experimental replicates.

Thank you. We have added densitometry quantitation of all western blots along with the statistical analyses to make it clear that all of the experiments were done as three independent sets and one set was presented as a representative blot in each case.

A major place for improvement would be to consider and discuss alternative explanations for the results, as outlined above in the public review.

Thank you for this important suggestion. We have now added alternative explanations of our results as suggested above.

It is a little misleading to label the signal for the HyPer probe "H2O2" (as is the case with all such probes). What is measured is the steady-state of a redox sensitive disulfide bond, which will be somewhat sensitive to H2O2 flux but also other factors such as the counteracting flux of reductants. It is a little unfair to refer to this as an H2O2 probe, and to DCF as a non-specific oxidant probe. While there is no such thing as a perfect ROS probe, and in general genetically encoded sensors offer a number of advantages, caution still needs to be exercised in interpreting results.

Thank you for this important suggestion. Now, we have changed the wordings from “H2O2 to HyPer fluorescence” in Figure 2 and Figure 2-figure supplement 2.

If I were to recommend another experiment (at the discretion of the editors and authors due to covid restrictions), it would be to measure NAD(P)H levels in the cells where antioxidants have been manipulated. Since many of these enzymes use NADPH as a substrate, changing their expression could impact the pools of these nucleotides, which could in turn impact redox sensitive processes. For example in Figure 3, the NADPH dependent GPX1 has an effect, but non-NADPH dependent antioxidants (MnSOD, CAT, PRDX3) do not.

Thank you for this important suggestion. Since we do not see any change in the hypoxic HIF1α stability or its transcriptional activity with the overexpression of another NAD(P)H dependent enzyme, GPX4, it did not provide us a strong rationale for NAD(P)H to be a critical mediator of the hypoxic HIF1α stability. Additionally, Non-NADPH dependent antioxidants such as catalase and PRDX3 led to an increase in the hypoxic HIF1α stability and its transcriptional activity while at the same time another Non-NADPH dependent antioxidant, MnSOD, did not have any effect on hypoxic HIF1α stability or its transcriptional activity. Overall, the effects of various antioxidants on the hypoxic HIF1α stability and/ or its transcriptional activity did not seem to be NAD(P)H dependent. We, therefore, could not think of measuring NAD(P)H levels in cells in response to antioxidant overexpression as a valuable set of critical experiment. We apologize for not doing this experiment.

Line 188 – text says NADPH decreases upon glucose ramp, but the signal in the figure (panel 1L) increases at this point.

Thank you for pointing this out. We have now corrected it.

Line 200 – "in" appears twice.

Thank you for pointing this out. We have now corrected it.

Line 266 – use of the term "reduce" to mean lower/decrease, should be rethought (same on line 298).

Thank you for this important suggestion. We have now replaced the word “reduced” with “decreased” throughout the manuscript.

The logic flow of the second paragraph of the paper (lines 70-88) is not strong. It starts out saying that one area where ROS signaling has been investigated is hypoxia. Then some information about the nuts and bolts of HIF signaling, and then something about diseases. Then the concluding line "understanding the role of ROS in HIF mediated adaptation has clinical implications". While yes, studying HIF is clinically relevant, the paragraph does not make a convincing case that studying ROS in the context of HIF signaling is relevant. Nothing in the middle makes a convincing case that any of the diseases associated with HIF are actually related to ROS. Some information on how ROS interplay with the HIF system is necessary, along with some details on how HIF is impacted at the molecular level in these diseases.

Thank you for this important suggestion. We have now improved the above mentioned paragraph in the manuscript as suggested.